# Evaluating the Wegener-Bergeron-Findeisen process in ICON in large-eddy mode with in situ observations from the CLOUDLAB project

Nadja Omanovic[1], Sylvaine Ferrachat[1], Christopher Fuchs[1], Jan Henneberger[1], Anna J. Miller[1], Kevin Ohneiser[2], Fabiola Ramelli[1], Patric Seifert[2], Robert Spirig[1], Huiying Zhang[1], and Ulrike Lohmann[1]

[1]Institute for Atmospheric and Climate Science, ETH Zürich, Zurich, Switzerland
[2]Leibniz Institute for Tropospheric Research (TROPOS), Leipzig, Germany

**Correspondence:** Nadja Omanovic (nadja.omanovic@env.ethz.ch) and Ulrike Lohmann (ulrike.lohmann@env.ethz.ch)

**Abstract.** The ice phase in clouds is essential for precipitation formation over continents. The underlying processes for ice growth are still poorly understood, leading to large uncertainties in precipitation forecasts and climate simulations. One crucial aspect is the Wegener-Bergeron-Findeisen (WBF) process, which describes the growth of ice crystals at the expense of cloud droplets leading to a partial or full glaciation of the cloud. In the CLOUDLAB project, we employ glaciogenic cloud seeding to initiate the ice phase in supercooled low-level clouds in Switzerland using uncrewed aerial vehicles with the goal to investigate the WBF process. An extensive set-up of ground-based remote sensing and balloon-borne in situ instrumentation allows us to observe the formation and subsequent growth of ice crystals in great detail. In this study, we compare the seeding signals observed in the field to those simulated using a numerical weather model in large-eddy mode (ICON-LEM). We first demonstrate the capability of the model to accurately simulate and reproduce the seeding experiments across different environmental conditions. Second, we investigate the WBF process in the model by comparing the simulated cloud droplet and ice crystal number concentration changes to in situ measurements. In the field experiments, simultaneous reductions in cloud droplet number concentrations with increased ice crystal number concentrations were observed with periods showing a full depletion of cloud droplets. The model can reproduce the observed ice crystal number concentrations most of the time, but not the observed fast reductions in cloud droplet number concentrations. Our detailed analysis shows that the WBF process appears to be less efficient in the model than in the field. In the model, exaggerated ice crystal number concentrations are required to produce comparable changes in cloud droplet number concentrations, highlighting the inefficiency of the WBF process in ICON.

## 1 Introduction

The ice phase is responsible for more than $70\%$ of precipitating clouds over continents and thus is essential for producing precipitation (Mülmenstädt et al., 2015; Heymsfield et al., 2020). The precipitation mainly originates from mixed-phase clouds, where ice crystals and cloud droplets coexist in an unstable thermodynamic equilibrium owing to subzero temperatures. In a mixed-phase cloud, three situations are possible: (i) both cloud droplets and ice crystals can grow if the ambient water vapor

pressure ($e$) exceeds saturation with respect to liquid water ($e_{s,w}$); (ii) ice crystals grow at the expense of evaporating cloud droplets if the ambient water vapor pressure $e$ lies between the saturation water vapor pressure with respect to ice and water ($e_{s,i} < e < e_{s,w}$); (iii) cloud droplets evaporate and ice crystals sublimate ($e < e_{s,w}$) (Korolev, 2007). The case of ice crystals growing at the expense of cloud droplets (ii) is called the Wegener-Bergeron-Findeisen (WBF) process, which is caused by the difference in water vapor supersaturation between the liquid and ice phase (Wegener, 1911; Bergeron, 1935; Findeisen, 1938). Whether $e$ exceeds $e_{s,w}$ or not depends, among other factors, on the vertical velocity as a source for water vapor, and on the integrated ice crystal surface (ice crystal number concentration × mean ice crystal radius), where the present ice crystals deplete the supersaturation by consuming the available water vapor generated by evaporating cloud droplets (Korolev and Mazin, 2003; Korolev, 2007). Here, Korolev and Mazin (2003) define the ice crystal radius as half of the maximum dimension of the particle. The ice crystals thereby speed up the formation of precipitation as they can rapidly grow to sizes where they sediment and induce collisions with other hydrometeors enabling further growth. Storelvmo and Tan (2015) showed the importance of the WBF process and how numerical models try to represent it within their grid scale. The parameterization of the WBF process directly impacts the representation of the liquid phase in mixed-phase clouds, which in turn impacts precipitation formation and patterns (Mülmenstädt et al., 2015; Heymsfield et al., 2020). This further influences the radiative responses of clouds (Xie et al., 2008). Currently, models can show a too-strong depletion of the liquid phase by the ice phase (Liu et al., 2011; McIlhattan et al., 2017; Huang et al., 2021) or a too-weak WBF process (Klaus et al., 2016; Kretzschmar et al., 2019) compared to observations. More studies focusing on the WBF process are therefore needed, which is the purpose of this work.

For the WBF process to take place in a cloud, ice crystals first need to form. They essentially follow two formation pathways: either via homogeneous (T < -38 °C) or heterogeneous nucleation (T > -38 °C), where the latter is essential for ice formation in mixed-phase clouds. Heterogeneous nucleation requires aerosols, so-called ice nucleating particles (INPs), to provide a surface for the ice to form on. These particles have numerous origins, with mineral dust being the most prominent natural INP, because it causes ice crystal nucleation over a wide range of temperatures (Hoose and Möhler, 2012; Murray et al., 2012; Ladino Moreno et al., 2013). The nucleation efficiency of INPs strongly decreases with increasing temperatures, where the lowest INP activity occurs close to the melting point of water. There, mainly fungal spores and bacteria are able to act as INPs, but due to their low abundance, there is generally a very low availability of INPs at temperatures close to 0 °C (Kanji et al., 2017).

This low INP availability at high temperatures can be exploited by glaciogenic cloud seeding to study the relevance of the WBF process. The general approach is to deliberately inject INPs into supercooled clouds to initiate ice formation and thus the subsequent growth of ice crystals to precipitation-sized particles (Haupt et al., 2018; Flossmann et al., 2019). The very first cloud seeding experiments date back to the 1940s and were conducted using dry ice or silver iodide (AgI) (Schaefer, 1946; Vonnegut, 1947). AgI serves as a particularly good INP with high ice activity at temperatures up to -5 °C due to its lattice structure which closely resembles that of ice (DeMott, 1995; Marcolli et al., 2016). This led to worldwide programs that actively pursued increasing precipitation over land to mitigate water scarcity (e.g. Woodley et al., 2003; Griffith et al., 2009; Geerts et al., 2010; Manton and Warren, 2011; Sin'kevich et al., 2018; Yang et al., 2018; Kulkarni et al., 2019; Wang

et al., 2019; Al Hosari et al., 2021; Benjamini et al., 2023). Often, wintertime orographic clouds are targeted for glaciogenic cloud seeding experiments, as the lifting of air along mountain slopes induces a high supercooled liquid water content, which serves as a water source for the ice crystals to grow (French et al., 2018; Tessendorf et al., 2019). Convective cells, which usually exhibit a larger content of supercooled water, are often too turbulent to successfully study the impact of cloud seeding with an observational setup (Flossmann et al., 2019). Such chaotic characteristics and the missing controlled reproducibility in field experiments obscure the feasibility of weather modifications (Haupt et al., 2018; Rauber et al., 2019). Only recently, the cloud seeding project SNOWIE was carried out to assess the impact of cloud seeding, from the release of seeding particles to the hydrometeor sedimentation with a focus on precipitation enhancement, and thus to study the whole microphysical process chain after initial ice formation and growth (French et al., 2018; Tessendorf et al., 2019). However, the limited observations of ice formation and growth processes active in the clouds further obscure the feasibility of glaciogenic cloud seeding (Flossmann et al., 2019).

Complementary to such field experiments, numerical models are employed to shed light on the statistical significance of cloud seeding by conducting repeated simulations in a controlled setup, which is not possible in a field experiment. Even though some processes are missing or simplified in models (Rauber et al., 2019; Morrison et al., 2020), they offer a way to quantify the impact of cloud seeding on a broader scale. One of the very first modelling studies on cloud seeding was conducted by Meyers et al. (1995) based on the laboratory studies of AgI ice nucleation activity from DeMott (1995). They reproduced field studies from the 1980s (Deshler et al., 1990) and successfully demonstrated an increase of precipitation in their seeded case. More recently, Xue et al. (2013a, b) investigated the seeding impact on orographic clouds and showed an increase in the effectiveness of cloud seeding if the INPs are injected directly into the cloud by airborne seeding as opposed to from the ground. Other modeling studies on convective clouds found similar results (e.g., Reisin et al., 1996; Ćurić et al., 2007; Chen and Xiao, 2010). More recent studies showed the importance of higher resolutions for evaluating the impact of cloud seeding. They employed a weather model in $100\,\mathrm{m}$ horizontal resolution, i.e. they conducted non-idealized large-eddy simulations (LES), and reproduced the environmental conditions and the dispersion of the seeding plume (Xue et al., 2016; Chu et al., 2017; Xue et al., 2022).

In this study, we evaluate the impact of cloud seeding in a numerical weather model (ICON, Zängl et al. (2015)) by utilizing seeding experiments conducted within the CLOUDLAB project. CLOUDLAB aims to improve our understanding of ice crystal formation and growth by exploiting the methodology of glaciogenic cloud seeding in dynamically stable clouds (Henneberger et al., 2023). Persistent low stratus clouds proved to be a good natural laboratory given their persistent and frequent occurrence over Switzerland during wintertime. To conduct the seeding experiments, an uncrewed aerial vehicle (UAV) attached with a seeding flare is flown into the cloud to release seeding particles (containing AgI) upstream of the field site. The particles are then advected downstream to the field site by the wind, where the seeding-induced microphysical changes (i.e., formation and growth of ice crystals) are observed by an extensive remote sensing setup together with in situ instrumentation at high spatio-temporal resolution (Henneberger et al., 2023; Miller et al., 2023). This approach allows for repeated and laboratory-like experiments in quick succession, offering the ideal opportunity to evaluate microphysical schemes in models. By improving the parameterizations of ice crystal growth in ICON with updated equations, CLOUDLAB aims to increase precipitation forecast

skills of numerical weather prediction models by first evaluating the ice crystal growth rate in seeded supercooled clouds in a high-resolution model.

Here, we present a series of LES using ICON (Zängl et al., 2015) focusing on the WBF process within the framework of CLOUDLAB. Section 2 introduces the observational setup in CLOUDLAB, our model setup including the new seeding parameterization, as well as the methods and data used for the analysis. To evaluate the WBF process within ICON, we first validated the ability of the model to reproduce the environmental conditions, such as temperature and cloud cover (Sect. 3.1). Afterwards, we conducted several seeding simulations to show the ability of the model to reproduce the seeding signal from selected field seeding experiments including the temperature dependency of the seeding parameterization and the dilution of the seeding plume (Sect. 3.2). Based on these simulations, we analyze the efficiency of the WBF process in the model compared to in situ observations, including the examination of the impact of the seeding particle emission rate on the WBF process (Sect. 3.3). We summarize our findings regarding the model performance with respect to simulating seeding experiments and ice crystal growth in Sect. 4.

## 2 Data and methods

### 2.1 CLOUDLAB: Observational setup

The CLOUDLAB project aims to improve the understanding of ice formation and growth through glaciogenic cloud seeding and entails three field campaigns: (1) January 2022 - March 2022, (2) December 2022 - February 2023, and (3) December 2023 - February 2024. During the first two campaigns, 55 successful seeding experiments were conducted at various temperatures, wind conditions, and seeding particle emission rates. Henneberger et al. (2023) details the observational setup including analysis of four seeding experiments of the first two campaigns. Miller et al. (2023) showed the possible seeding patterns conducted in the field experiments. Their Fig. 7 displays how in-cloud experiments consist of several seeding legs per experiment, where the UAV is flying back and forth while the seeding flare is burning for approximately 6 min. The field experiments discussed in this study were performed with 4 legs of 400 m each. Here, we focus on five seeding experiments which were conducted on two consecutive days: Three experiments on 25 January 2023 (S25-2, S25-2.5, S25-3), which are also discussed in Henneberger et al. (2023), and two experiments on 26 January 2023 (S26-2.5a, S26-2.5b). We closely followed the seeding distances from the field site, the seeding patterns, and the seeding start time (Table 1). The seeding simulations are named by combining their experiment date and their seeding distance, e.g. the seeding experiment on 25 January 2023 at 2.5 km distance is called S25-2.5. For 26 January 2023, where both experiments are identical in their setup, we introduced an additional identifier in the form of "a" and "b", i.e. S26-2.5a and S26-2.5b. These two identical setups serve as a test for the validity of the signal we observe in the field experiments, but also in the model simulations.

Both January days were characterized by low stratus clouds and low temperatures, which are the targeted conditions for conducting seeding experiments in this project. Here, we only briefly discuss the relevant instrumentation, and a full description of the experimental approach and instrumentation can be found in Henneberger et al. (2023).

For model validation, we compared the model simulations to the atmospheric profiles measured by radiosondes (Sparv S1H3,
Windsond) and to the radar reflectivity observed by a vertically pointing cloud radar (FMCW-94-DP, Radiometer Physics
GmbH) (Sect. 3.1). The seeding simulations (Sect. 3.2) are compared to elevation scans conducted by a cloud radar (Mira-35,
Metek) and in situ measurements obtained with a tethered balloon system (TBS) (Sect. 3.2.2 and 3.3). The TBS is equipped
with a holographic imager for microscopic objects (HOLIMO) to observe cloud characteristics, such as cloud droplet and ice
crystal number concentrations and size distributions. HOLIMO measures particle diameters in the range between $6\,\mu m$ and
$2\,mm$ (Ramelli et al., 2020). Given the non-spherical shape of ice crystals, a mean area equivalent radius is calculated for a
more direct comparison to the model data in Fig. 10. The differentiation between cloud droplets and ice crystals is based on
the particle's shape for radii larger than $25\,\mu m$. Anything below that is identified as cloud droplets due to the resolution limit of
HOLIMO. After applying a neural network (Touloupas et al., 2020) to classify the particles into cloud droplets and ice crystals,
a manual labeling of all ice crystals was conducted to minimize misclassifications. Overall following uncertainties apply: cloud
droplet number concentration ($\pm\,5\,\%$), ice crystal number concentrations for particles larger than $100\,\mu m$ in diameter (5-10 %),
ice crystal number concentrations for particles smaller than $100\,\mu m$ in diameter (15 %) (for further information see Beck (2017)
and Ramelli et al. (2021)). The HOLIMO data were analyzed with a $5\,Hz$ frequency during the seeding experiment, whereas
the background cloud was analyzed with $1\,Hz$. Subsequently, the HOLIMO data were averaged over 5 data points over time,
resulting in $1\,s$ averages for the seeding signal and $5\,s$ for the background. The time periods before and after seeding are used
to observe the unperturbed background characteristics of the cloud. We further assume that the highly localized measurements
by HOLIMO serve as a representative sample of the cloud characteristics inside the seeding plume and of the background due
to persistent low stratus clouds in a stable boundary layer.

## 2.2 Model and simulation description

We employed the numerical weather prediction model ICON-v2.6.5 (Zängl et al., 2015) in a one-way nested mode (Fig. 1).
The outermost nest has a $1\,km$ horizontal resolution, 80 vertical levels (up to 22 km), a model time step of $10\,s$, and is based on
the operational grid of the Swiss National Meteorological Service (MeteoSwiss) (Schmidli et al., 2018). The second nest with
$260\,m$ horizontal resolution and a model time step of $2.5\,s$ is located over the Swiss plateau, centered around the field site. The
final nest is of $130\,m$ horizontal resolution with $1\,s$ model time step and a domain size of $40\,x\,30\,km^2$ (Fig. 1). Both inner nests
also have 80 vertical levels and a vertical grid spacing ranging from approximately $20\,m$ (at cloud base) to $80\,m$ (at cloud top).
The initial and boundary conditions for the outermost nest are based on hourly COSMO analysis data generated by MeteoSwiss,
and the simulations were initialized at 00 UTC and performed for 23 hours. For each simulation, we conducted a reference (no
seeding) and a seeding simulation to quantify the impact of cloud seeding. The innermost nest with the highest resolution is
the focus of this analysis. All three nests employ the ecRad scheme (Hogan and Bozzo, 2018), the "3D Smagorinsky diffusion"
turbulence scheme (Lilly, 1962; Smagorinsky, 1963; Dipankar et al., 2015), and the Seifert and Beheng (2006) two-moment
microphysics scheme. The cloud condensation nuclei concentration was set to $1000\,cm^{-3}$ following Schmale et al. (2018) for
rural and continental areas (Melpitz, Germany; CESAR tower, the Netherlands; and Vavihill, Sweden) during wintertime and is
uniformly distributed in the domain. The frequency of model output was set to $5\,min$ after also testing $1\,min$ output frequency,

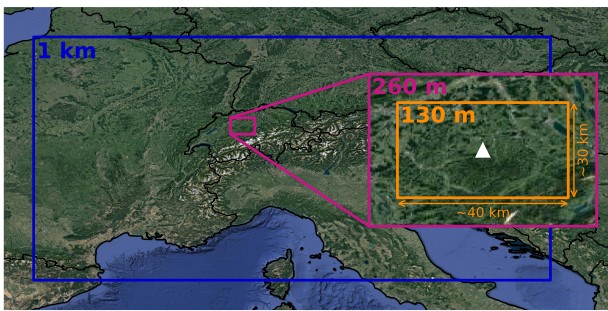

**Figure 1.** Nesting setup with three nests from 1 km (blue, MeteoSwiss setup), 260 m (pink) down to 130 m (orange). The CLOUDLAB field site is marked with a white triangle. The two last domains were chosen such that the field site is located in the center. Map taken from Google satellite images (©Google Maps).

which showed similar results as in the 5 min output. Moreover, calculating the expected arrival time of the seeding plume at the field site (seeding start and growth time, see Table 1) shows that the expected arrival and a full 5 min model output timestep are very close (within ± 1 min).

In the following we provide a short description of the two-moment microphysics scheme used in the model. For more details, please refer to Seifert and Beheng (2006). The scheme tracks mass and number mixing ratios for six hydrometeors: cloud droplets, ice crystals, snowflakes, graupel, hail, and rain drops by assuming a Gamma-distributions for the underlying size distributions. All relevant cloud processes are parameterized, such as cloud droplet activation, ice crystal nucleation, growth of ice crystals by water vapor deposition (i.e., the WBF process), riming, melting, and sublimation of ice crystals. The maximum diameter and terminal fall velocity are parameterized following power laws with constant coefficients (Seifert and Beheng, 2006). As we are investigating the ice processes within mixed-phase clouds, we provide additional information relevant to ice particles. The ice crystal shape is set to be spherical, which is a simplification in the scheme given the large variety of shapes (Bailey and Hallett, 2009). In this study, we do not change the shape of the ice crystals as we want to investigate the ice crystal growth rate in the default configuration of the model. During the conducted seeding experiments, we mostly measured needles or columns. When we compare the ice crystal sizes in Fig. 10, we investigate the mean equivalent radius of ice crystals.

Simulations S26-2.5a and S26-2.5b are identical in their setup and serve as a proof-of-concept for the seeding approach in the model. The simulations S25-2, S25-2.5, and S25-3 highlight the impact of the seeding distance on the ice crystal growth time. These three simulations have been conducted at a warmer temperature than S26-2.5a and S26-2.5b. We use this difference in environmental conditions to evaluate the temperature dependence of ice nucleation induced by the seeding particles.

At subzero temperatures secondary ice production can occur, which is also parameterized in the model. For secondary ice production to occur in the model rimed graupel particles are needed, but their concentrations are close to zero in the model; hence we can exclude the effect of secondary ice production in our analysis. During the field experiments, we also expect a low secondary ice production rate given that only a few larger cloud droplets (with radii > 20 µm) are present. Riming on the ice

**Table 1.** Overview of the performed seeding simulations showing the CLOUDLAB ID (ID), the simulation name given in this study (Name), the distance of the seeding location to the field site, the seeding start time (Start), the seeding height (Height) conducted in the field (Obs) and in the model (Model), the observed (Obs) and simulated (Model) temperature, wind speed and wind direction, and the growth time. Observed temperature measurements are taken from the UAV. The wind speed is calculated based on the time between seeding initiation and the first signal detected in the vertically pointing radar, and the wind direction is taken from the TBS. Based on the seeding distance and wind speed calculations, a growth time is determined assuming immediate ice nucleation of the seeding particles. Note that the last three experiments were actually performed on 25 January 2023, but due to a mismatch between model and observation temperature, still simulated on 26 January 2023 at a lower height to match the seeding temperature.

| ID | Name | Distance (km) | Date | Start (UTC) | Height (km AMSL) | | Temperature (°C) | | Wind speed (ms$^{-1}$) | | Wind direction (°) | | Growth time (min) |
|---|---|---|---|---|---|---|---|---|---|---|---|---|---|
| | | | | | Obs | Model | Obs | Model | Obs | Model | Obs | Model | |
| SM063 | S26-2.5a | 2.5 | 26.01. | 10:22 | 1.35 | 1.34 | -6.4 | -6.5 | 5.2 | 4.8 | 67 | 79 | 8.0 |
| SM064 | S26-2.5b | 2.5 | 26.01. | 10:48 | 1.35 | 1.34 | -6.2 | -6.5 | 5.8 | 5.4 | 68 | 80 | 7.1 |
| SM059 | S25-2 | 2.0 | 25.01. | 10:50 | 1.30 | 1.20 | -5.4 | -5.6 | 5.5 | 5.5 | 82 | 77 | 6.1 |
| SM058 | S25-2.5 | 2.5 | 25.01. | 10:28 | 1.30 | 1.20 | -5.5 | -5.6 | 5.2 | 4.8 | 81 | 68 | 8.0 |
| SM060 | S25-3 | 3.0 | 25.01. | 11:15 | 1.30 | 1.20 | -5.4 | -5.6 | 5.5 | 4.8 | 89 | 74 | 9.1 |

180 crystals was also only visible in two out of the five experiments (S26-2.5a/b). In addition, if splinters occurred, they probably did not grow large enough to be detected given the short growth time during the experiments (see Table 1).

The selection of the presented seeding simulations was constrained by how accurately the model reproduced the observed environmental conditions. Unfortunately, the model overestimated the temperatures for 25 January 2023 (Henneberger et al., 2023) (Fig. 3a), while the temperatures on the 26 January 2023 were simulated adequately (Fig. 3b). For this reason, we

185 decided to utilize the simulation of 26 January 2023 for all seeding experiments conducted on 25 and 26 January 2023 (see also Sect. 3.1) given the presence of persistent low stratus clouds with north-easterly to easterly winds on both days. While the experiments on 26 January 2023 (S26-2.5a, S26-2.5b) were simulated at the same seeding height as the field experiments (Fig. 4), the experiments on 25 January 2023 (S25-2, S25-2.5, S25-3) were simulated at a lower height such that the model temperature matches the seeding temperature from the field (-5.5 °C, Table 1).

190 ## 2.3 Seeding parameterization and seeding setup

To simulate glaciogenic cloud seeding in ICON, we implemented a deterministic freezing parameterization specifically for the seeding particles (AgI) used in the field. Henneberger et al. (2023) hypothesized that the seeding particles are highly hygroscopic and postulated that these either are taken up by existing cloud droplets or undergo cloud droplet activation before freezing. Furthermore, their clear-sky seeding experiments showed that the seeding particles have a mean particle diameter

195 between 100 and 400 nm. The freezing ability of AgI particles for sizes between 20 and 400 nm was measured by Marcolli

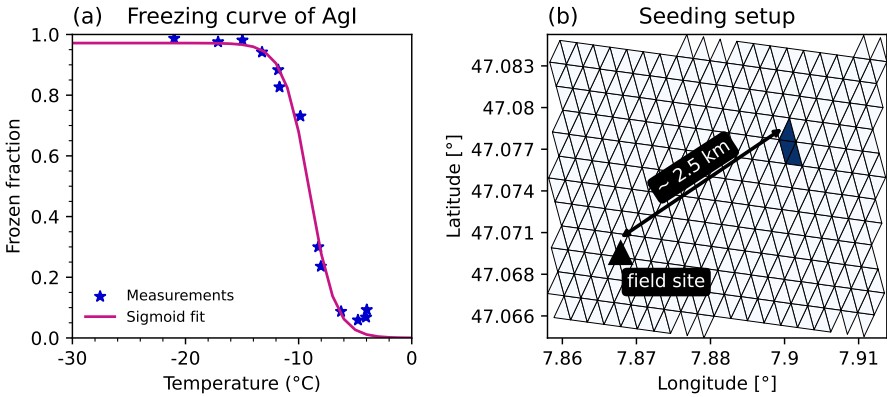

**Figure 2. (a)** Dependence of frozen fraction of AgI particles on temperature. Measurements (blue stars) were taken from Marcolli et al. (2016) for particle diameters of 400 nm. The pink line shows a sigmoid fit through the measurements. **(b)** Seeding setup for the S26-2.5a and S26-2.5b experiments in terms of location and distance from the field site, and with a seeding particle emission rate of $10^6$ m$^{-3}$s$^{-1}$ (marked in dark blue). The triangular grid represents the native grid of ICON.

et al. (2016) in a laboratory setup (their Fig. 1) showing a strong size and temperature dependence for the frozen fraction (FF). However, for sizes larger than 40 nm, the freezing curves are fairly similar, thus we decided to follow the 400 nm measurements. We obtained the relationship

$$FF = \frac{-b}{1 + \exp(-k(T - T_0))} + b, \tag{1}$$

with the parameters $b = 0.97$, $k = 0.88$, $T_0 = 263.95$ K, and $T$ being the temperature (K). Figure 2a shows the laboratory measurements (400 nm) of Marcolli et al. (2016), indicating an increase in frozen fraction with decreasing temperature, highlighting the strong temperature dependence of the ice activity of AgI particles. Within our model setup, we introduced the Marcolli parameterization in the two-moment microphysics scheme before the dust freezing parameterization takes place (Seifert and Beheng, 2006). For the analysis, we assume immediate ice nucleation given the high ice nucleation activity of AgI

below -5 °C. Further, we limit the ice nucleating activity by the availability of cloud droplets. Hence, the number of ice crystals cannot exceed the number of available cloud droplets.

In the model, the seeding particles were introduced along a 400 m leg to mimic the seeding pattern conducted in the field experiments (see Sect. 2.2). We used the coordinates of the seeding legs to define the injection area for the seeding particles, as shown in Fig. 2b. Each conducted simulation received the same amount of seeding particle emission rate: $10^6$ m$^{-3}$s$^{-1}$

seeding particles were released in three model grid boxes for 6 min corresponding to the burning time of one seeding flare. This seeding particle emission rate is based on a series of sensitivity simulations for seeding experiment S26-2.5a, where we injected different concentrations of seeding particles into the model and compared the simulated ice crystal number concentrations to the observations. The seeding particle emission rate and thus the seeding setup were constrained by the ice crystal number concentrations observed by HOLIMO and chosen in such a way that they match the seeding simulation S26-2.5a (Sect. 3.3.1).

## 2.4 Detection of seeding plume

We applied a simple method to extract the seeding signal from the background. We took the difference in ice crystal number concentrations between a seeding simulation and a reference simulation (no seeding) to remove the background and isolate the seeding plume. The seeding plume was then defined by a threshold ice crystal number concentration of $0.001\,\mathrm{cm}^{-3}$. We used the identified seeding plume as a mask for extracting further quantities in the seeding simulation, but also in the difference between the seeding and reference simulation, such as cloud droplet number concentrations, temperature, and updraft changes caused by the seeding perturbation. The analysis of the seeding plume and related quantities is based on this approach to quantify the cloud seeding impact. For the comparison to in situ measurements, we considered the model output time step closest to the expected arrival of the seeding signal at the field site using the calculated growth time in Table 1. For the simulations S25-2, S25-2.5, and S25-3 the growth times differ from the ones reported in Henneberger et al. (2023), because here we use the time between the ignition of the seeding flare and the first increase in radar reflectivity in a vertically pointing radar at the field site (i.e. arrival of seeding plume). Henneberger et al. (2023) used the wind profiler for the wind measurement and estimated the advection time from this. However, the lowest level of measurements in the wind profiler was consistently higher than the actual seeding altitude, which leads to an overestimation of the wind speed, and thus an undererstimation of the ice crystal growth time.

## 3 Results

### 3.1 Model validation

To validate the model, we compared the simulated to the observed conditions on 25 and 26 January 2023. Figure 3 shows the vertical profiles of temperature and relative humidity measured by a radiosonde (solid lines) launched from the field site and predicted by the model (dashed and dotted lines). Both days were characterized by low level clouds with subzero temperatures, with 26 January showing a deeper cloud and colder temperatures. In both cases, the model did not reproduce the sharp inversions at cloud top, and for 25 January 2023, it did not simulate cold enough temperatures for the seeding particles to be effective. Therefore, we used the 26 January 2023 simulation as a surrogate for all seeding simulations (see also Sect. 2.2), as it better represents the observations on 25 January 2023 (Fig. 3a). The morning of 26 January 2023 was characterized by a strong temperature inversion at around $1.6\,\mathrm{km}$ above mean sea level (AMSL) limiting cloud top to that height. The model predicted a lower and weaker inversion compared to the observations with a corresponding lower cloud top. This discrepancy does not pose a problem as the observed and simulated temperatures at the seeding height (see temperature in Table 1) are in good agreement, which is the strongest constraint for our seeding simulations. Regarding the wind speed, the model performs reasonably well with slight underestimations for four of the cases ($0.4$ - $0.7\,\mathrm{m\,s}^{-1}$, Table 1). However, the model performs worse with regard to wind direction ($\pm\,13°$, Table 1). The discrepancy in wind speed and direction does not impact the ability of the model to simulate seeding experiments.

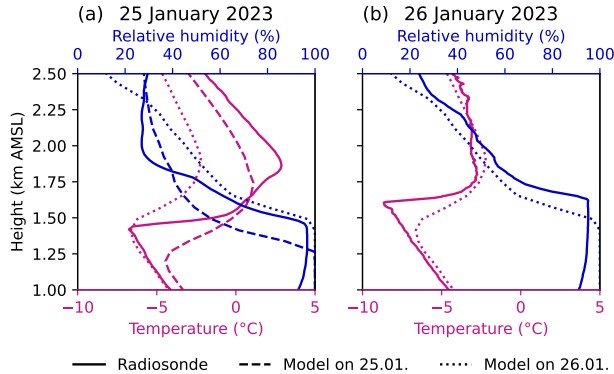

**Figure 3.** Comparison of radiosonde (solid lines) and model (dashed and dotted lines) profiles of the lower atmosphere for temperature (°C, pink) and relative humidity (%, blue). The simulated profiles on 25 January 2023 12:00 UTC (dashed) and 26 January 2023 10:00 UTC (dotted) are averaged over four grid cells around the field site. **(a)** shows the radiosonde launch at 25 January 2023 12:08 UTC with the model output from both days, while **(b)** shows the radiosonde launch at 26 January 2023 09:29 UTC with the model output on 26 January.

In addition, we compared the observed and predicted cloud cover at the field site by taking the radar reflectivity of a vertically pointing radar as a proxy for cloud cover and the computed cloud cover from the prognostic cloud water mass in the model (Fig. 4). We see that the model predicts a long-lasting low cloud that reaches slightly lower cloud top heights than observed by the radar with the seeding simulations still being fully inside the cloud. The lower cloud top can be also seen in the comparison
of relative humidity in the radiosonde profiles.

### 3.2 Ice response due to seeding

In the following, we discuss the seeding impact on cloud droplet and ice crystal number concentrations. We first show the evolution of the seeding plume in terms of time and location, and compare it to radar scans (Sect. 3.2.1). We then compare the observed and predicted ice crystal number concentrations for all simulations (Sect. 3.2.2). This is followed by the investigation
of the WBF process in the model by utilizing the in situ measurements (Sect. 3.3). We conclude the result sections with the impact of the seeding particle emission rate on ice crystal number concentration and cloud droplet reductions (Sect. 3.3.1).

#### 3.2.1 Evolution of the seeding plume

Here we examine the evolution of the seeding plume with respect to changes in ice crystal number concentrations in the seeding simulation S26-2.5a. The seeding plume tracks for the other simulations are in Appendix A. Figure 5 shows the response in
ice crystal number concentrations taken at the level of maximum concentration at each model output time step (top view) and as a cross section along the mean wind direction at each model output time step. The first output time step ($t_1$) denotes the first five-min interval after the seeding start (here 10:25 UTC). We observe a strong and sudden increase in ice crystal number concentrations of up to $1\,\mathrm{cm}^{-3}$ (see Table 2) that is rapidly diluted within the first 15 min as the seeding plume is advected

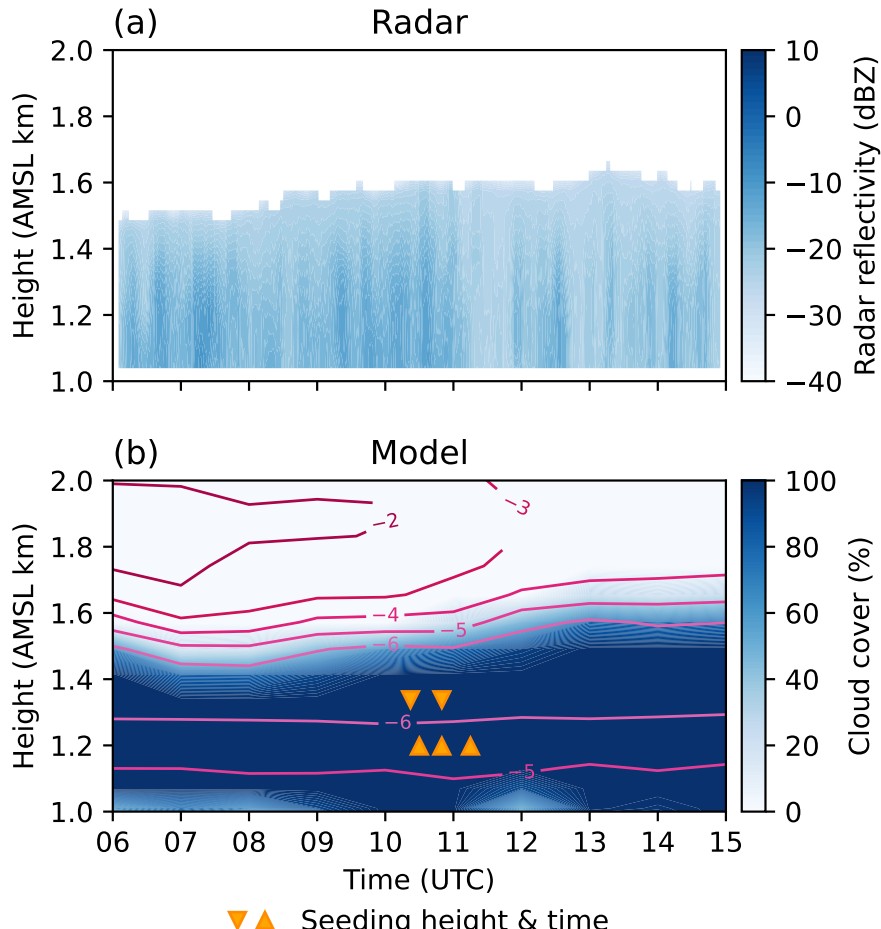

**Figure 4.** Temporal evolution of the measured radar reflectivity (dBZ, **(a)**) and the simulated cloud cover (%, **(b)**) at the field site as a function of height using the radar reflectivity as a proxy for cloud extent. The radar reflectivity was averaged over 5 min intervals. **(b)** in addition shows the model temperature profile (°C, pink isotherms) and the seeding simulations in terms of height and time (orange triangles). Upward facing triangles represent the simulations S25-2, S25-2.5, and S25-3, and downward facing triangles the simulation S26-2.5a and S26-2.5b.

along the main wind direction. The plume also spreads out horizontally starting from several hundred meters ($t_1$ and $t_2$) to about 2 x 3 km² at $t_7$. The seeding plume almost missed the field site (black triangle) which can be attributed to the mismatch in wind directions between the model and observations (Table 1). The seeding plume not only spreads out horizontally, but also vertically due to turbulence and orographic lifting as shown in Fig. 5b, where we observe a vertical extent of up to 500 m. We also see the sedimentation of the ice crystals as the seeding plume descends in height. Based on the applied detection method we identified the seeding plume for approximately 35 min before it vanished.

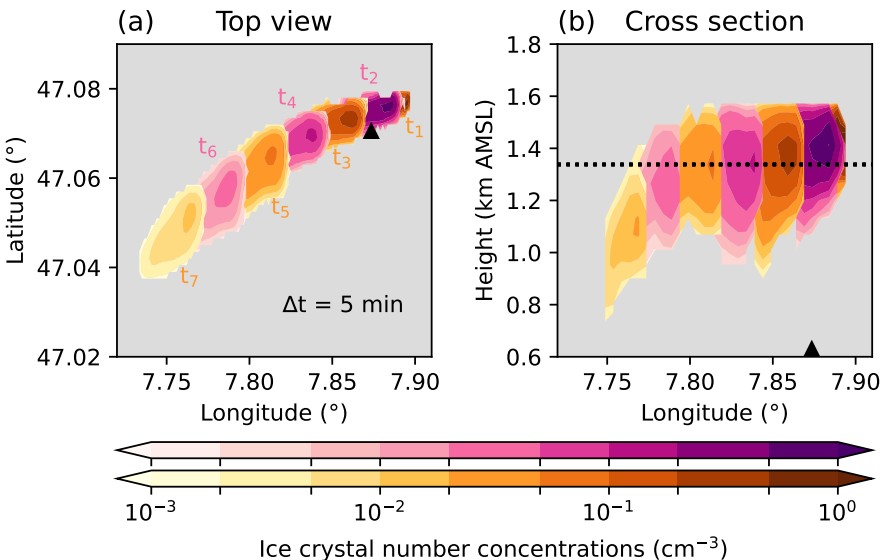

**Figure 5.** Simulated ice crystal number concentration changes $(\mathrm{cm}^{-3}$, colormap) after seeding in simulation S26-2.5a with **(a)** showing the top view at the level of maximum ice crystal concentration for each model output time step. $t_1$ is the first five-min output time step after the seeding start, i.e. here 10:25 UTC. Alternating colors are used for better visibility between the output time steps. **(b)** shows the vertical cross section of the ice crystal number concentration $(\mathrm{cm}^{-3})$ along the mean wind direction at the seeding height (dashed line in **(b)**) for each five-min output time step. The black triangle denotes the field site in both panels.

To evaluate the horizontal and vertical extent of the simulated seeding plume, we used elevation scans that were conducted by a scanning radar during the field experiment. The radar performed repeated elevation scans in the plane perpendicular to the wind direction (from north-east), thus allowing us to observe the horizontal and vertical extent of the seeding signal. Figure 6 shows the observed and simulated radar reflectivity (dBZ) at 10:30 UTC on 26 January 2023. The simulated radar reflectivity is based on an implemented Rayleigh approximation for the backscattering of the cloud particles, where for frozen hydrometeors

it is differentiated between dry and wet ice, snow, and graupel. While we can identify more fine-granular structures in the radar observation, the simulated radar reflectivity also shows an increase in reflectivity inside the seeding plume and a vertical spreading out throughout the cloud layer for the same time (10:30 UTC).

### 3.2.2    Observed vs. predicted ice crystal number concentrations

     In Fig. 7, we compare the observed and predicted ice crystal number concentrations as frequency distributions for the exper-

iments S26-2.5a and S26-2.5b. The observations were taken by the in situ instrument HOLIMO aboard the TBS over a time interval of $10\,\mathrm{min}$. For both data sets, we set an ice crystal number concentration threshold of $0.001\,\mathrm{cm}^{-3}$. Model predictions are based on the identified seeding plume at the time of the expected arrival of the seeding signal (see growth time in Table 1). The comparison of the measured and simulated radii is shown in Fig. 10p and discussed in Sect. 3.3.

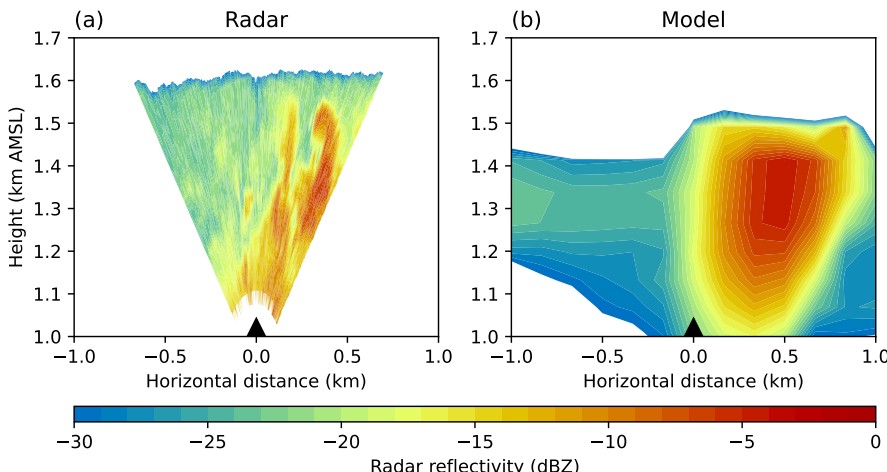

**Figure 6.** Comparison of the radar reflectivity measured by a scanning cloud radar with a scanning frequency of 90 s per scan (Mira-35, Metek, **(a)**) and simulated by the model (**(b)**)) during seeding experiment S26-2.5a at 10:30 UTC (i.e. model output time step $t_1$). The field site is located in the center of the cross section (black triangle, 0.0). Both figures show a cross section of the seeding signal perpendicular to the main wind direction.

In Fig. 7a, we see that the maximum observed and simulated ice crystal number concentrations are in good agreement,
with the model showing slightly higher concentrations (see Table 2). For experiment S26-2.5b, which has an identical setup as S26-2.5a, a similar pattern is observed (see Fig. 7b), emphasizing that the model configuration can be used to conduct seeding experiments for further investigation of the ice crystal growth inside the cloud. However, in both simulations, the median and mean concentrations are strongly underestimated, which is further discussed below (Sect. 3.3). The seeding particle emission rate ($10^6$ seeding particles $\mathrm{m}^{-3}\mathrm{s}^{-1}$) used in this study is probably an upper estimate given that the surrounding of the seeding
plume in general is warmer in the model (i.e., higher temperatures below the inversion) than observed which leads to a lower activation rate of INPs compared to reality. Hence, we need to introduce more seeding particles to achieve the same ice crystal number concentrations as observed.

The responses in ice crystal number concentrations for the seeding simulations S25-2, S25-2.5, and S25-3, which were conducted at different seeding distances from the field site, are shown in Fig. 8. The observations indicate that with increasing
seeding distance, the measured ice crystal number concentrations are reduced, indicating a dilution effect of the seeding signal. In experiment S25-2 (closest distance), we measured ice crystal number concentrations of about 2.5 $\mathrm{cm}^{-3}$, while experiment S25-3 (farthest distance) only measured concentrations of around 0.5 $\mathrm{cm}^{-3}$. The model, however, fails to reproduce the very high concentrations of S25-2, which may be due to an underestimation of the ice nucleation activity of the seeding particles at temperatures close to -5 °C. Also, the ice crystal nucleation rate is constrained by the available cloud droplet number
concentrations, which were underestimated in the model compared to the observations (Fig. B1). An additional reason could be the aerosol concentration, which was adapted to the simulation S26-2.5a. Hence, we cannot simulate the highest observed ice

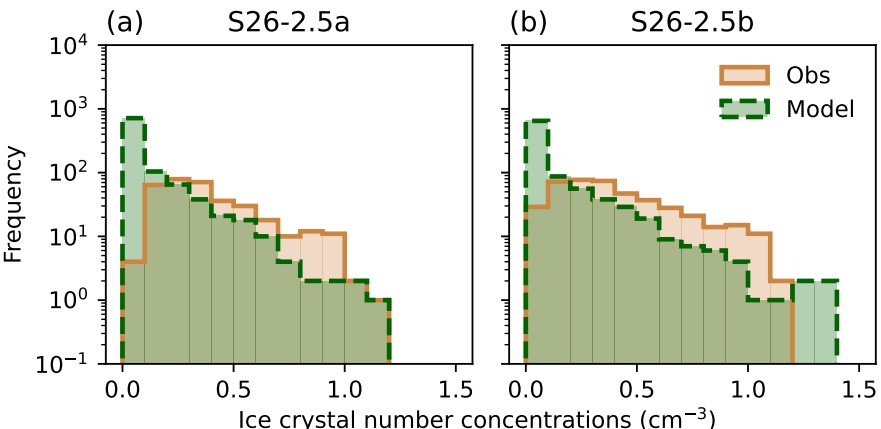

**Figure 7.** Frequency distributions of the observed (brown, solid) and predicted (green, dashed) ice crystal number concentrations ($\mathrm{cm}^{-3}$) for experiments S26-2.5a (**(a)**) and S26-2.5b (**(b)**). The bin size is $0.1\,\mathrm{cm}^{-3}$.

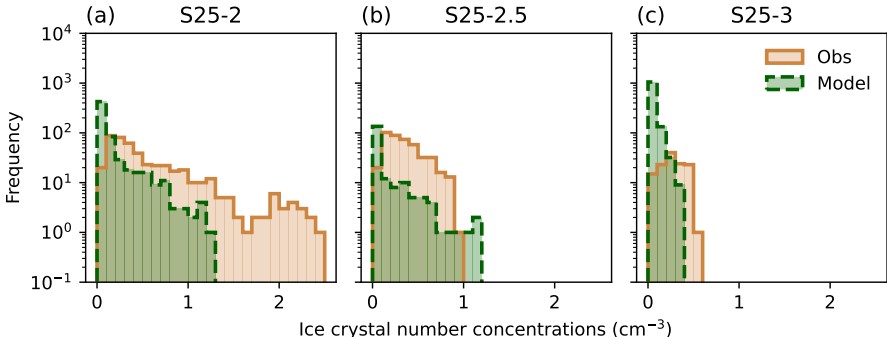

**Figure 8.** Frequency distributions of the observed (brown, solid) and predicted (green, dashed) ice crystal number concentrations ($\mathrm{cm}^{-3}$) for experiments S25-2 (**(a)**), S25-2.5 (**(b)**), and S25-3 (**(c)**). The bin size is $0.1\,\mathrm{cm}^{-3}$.

crystal number concentrations. In simulation S25-2.5, which has an identical setup as the S26-2.5a and S26-2.5b simulations except for being at $1\,°\mathrm{C}$ warmer seeding temperatures, we can see that the ice crystal number concentrations are slightly overestimated compared to the observations (Fig. 8b), which is comparable to the results from the simulation of S26-2.5b. For simulation S25-3, the model slightly underestimates the ice crystal number concentrations. This is in accordance with an efficient dilution in the model as the ice crystal number concentrations rapidly decrease with time, as shown in the seeding plume track analysis (Fig. A1).

### 3.3 Investigating the Wegener-Bergeron-Findeisen process

The in situ observations taken during the seeding experiments allow us to investigate the WBF process in greater detail. Figure 9a shows the measured cloud droplet number concentrations and Fig. 9b the ice crystal number concentrations during the seeding experiment S26-2.5a, including measurements of the undisturbed background cloud prior to and after the seeding. The analysis for the other experiments is shown in Appendix C (Fig. C1) and an overview of all experiments is given in Table 2. The background cloud had an ice crystal number concentration of $0\,\mathrm{cm}^{-3}$ and a median cloud droplet number concentration of $320\,\mathrm{cm}^{-3}$ (Fig. 9). During the seeding experiment, the ice crystal number concentrations increase up to $1\,\mathrm{cm}^{-3}$, whereas the cloud droplet number concentrations simultaneously decrease by up to $300\,\mathrm{cm}^{-3}$.

To quantify the reduction in cloud droplets throughout the seeding experiment, we first defined the cloud droplet number concentration of the background state by calculating the median over a $20\,\mathrm{min}$ period of the background. Second, we considered all times in our analysis where the observed ice crystal number concentration was larger than $0.001\,\mathrm{cm}^{-3}$. The time periods considered in the analysis are marked by the vertical brown lines in Fig. 9a/b and the relative reductions in the cloud droplet number concentrations with regard to the median concentration are shown as a frequency distribution in Fig. 9c-e for different model output time steps. The observations indicate that between the seeding start and the measurements (i.e., $8\,\mathrm{min}$) the liquid phase was entirely depleted during some time periods (i.e., cloud droplet reductions of $100\,\%$), emphasizing the high efficiency of the WBF process observed in the field. Thus, the freshly nucleated ice crystals are highly efficient in consuming water vapor from the evaporating cloud droplets, leading to increased ice crystal growth and eventually isolated glaciation patches inside the cloud. These strong reductions in cloud droplets can partly originate from riming, where the cloud droplets immediately freeze onto the ice crystals. Only for experiments S26-2.5a/b riming was observed with HOLIMO. Here, we solely focus on the WBF process but we are aware that also riming can lead to additional reductions in the liquid phase.

Next, we compare the observations to the simulated changes in the model by taking the model output time step that is closest to the time the seeding signal is expected to arrive at the field site. The changes in the model are computed from the difference between the reference and seeding simulation. In addition, we constrained the seeding plume by the available liquid water content inside the cloud: We only considered grid cells in the analysis where the ice crystal number concentration is larger than $0.001\,\mathrm{cm}^{-3}$ and the liquid water content larger than $0.1\,\mathrm{g\,m}^{-3}$. This way we only include grid cells where the WBF process actually could take place. Figure 9c shows both the observations and model response and we can see that the model is not able to reproduce such strong reductions in cloud droplet number concentrations, indicating that the ice crystal growth in the model is underestimated. This discrepancy may originate from the computation of the ventilation coefficient, which determines the speeding up of the diffusional growth due to turbulent motions. This hypothesis will be investigated in future studies. When looking at later model output time steps, we see that the model predicts a comparable reduction in cloud droplet number concentrations as observed (Fig. 9d-e). This further points to the fact that the WBF process in the model in its current form is not efficient enough as more time is needed to reach similar cloud droplet reductions.

In Table 2, we show the median, mean, and maximum ice crystal number concentrations (absolute values). For cloud droplets, we report the reduction of the median, mean, and maximum cloud droplet number concentration relative to the undisturbed

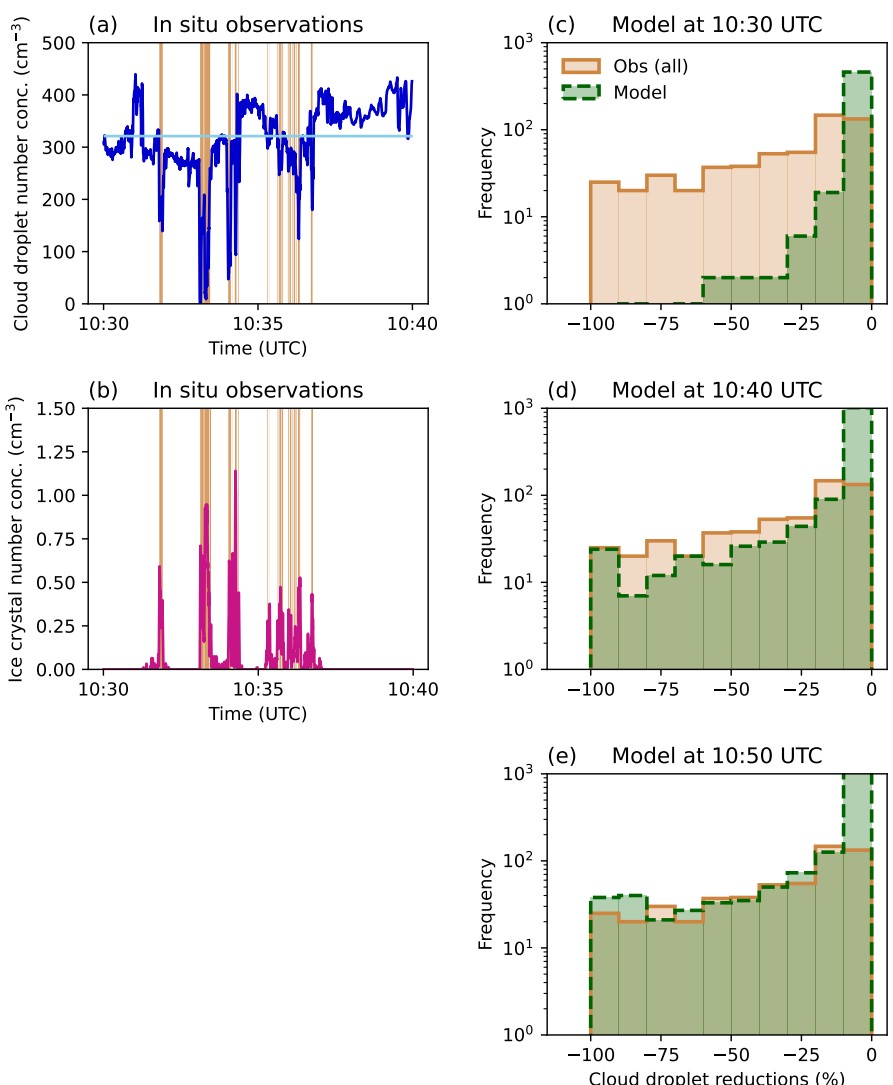

**Figure 9. (a)** Cloud droplet ($\mathrm{cm}^{-3}$, blue) and **(b)** ice crystal number concentrations ($\mathrm{cm}^{-3}$, pink) measured by HOLIMO during the seeding experiment S26-2.5a. The horizontal light blue line in **(a)** shows the median cloud droplet number concentration over a $20\,\mathrm{min}$ time span of background cloud prior/after the seeding. The brown shading indicates time periods where the ice crystal number concentration was above $0.001\,\mathrm{cm}^{-3}$, and which were considered for the cloud droplet reduction analysis. The frequency distributions for the observed (brown, solid) and simulated (green, dashed) cloud droplet reductions are shown in **(c)-(e)** for different model output time steps. The model output time steps are 10:30 UTC, 10:40 UTC, and 10:50 UTC for the panels **(c)-(d)**, respectively, and correspond to the plumes in $t_2$, $t_4$, and $t_6$ in Fig. 5 and Fig. 10. The bin size is set to $10\,\%$.

background (relative values) to account for the lower median cloud droplet number concentration of approximately $100\,\mathrm{cm}^{-3}$ in the model compared to the observations (Fig. B1). The maximum ice crystal number concentrations are in good agreement

**Table 2.** Median, mean, and maximum observed (Obs) and predicted (Model) ice crystal number concentrations $(\mathrm{cm}^{-3})$ and cloud droplet reductions (%) for all five experiments. For the observations we considered the entire measurement period during the seeding experiment. For the model we considered the output time step that is closest to the time of the expected seeding signal at the field site, i.e. based on the calculated growth time in Table 1.

| | Ice crystal number concentration $(\mathrm{cm}^{-3})$ | | | | | | Cloud droplet reduction (%) | | | | | |
| | Median | | Mean | | Maximum | | Median | | Mean | | Maximum | |
| Name | Obs | Model | Obs | Model | Obs | Model | Obs | Model | Obs | Model | Obs | Model |
|---|---|---|---|---|---|---|---|---|---|---|---|---|
| S26-2.5a | 0.2 | 0.02 | 0.27 | 0.10 | 1.0 | 1.2 | -20 | -1.75 | -30 | -1.75 | -100 | -80 |
| S26-2.5b | 0.2 | 0.02 | 0.30 | 0.11 | 1.2 | 1.3 | -20 | -0.12 | -25 | -0.5 | -100 | -15 |
| S25-2 | 0.25 | 0.25 | 0.40 | 0.12 | 2.4 | 1.25 | -35 | -0.003 | -40 | -0.03 | -100 | -0.4 |
| S25-2.5 | 0.15 | 0.03 | 0.26 | 0.12 | 0.9 | 1.2 | -30 | -0.015 | -35 | -0.0001 | -100 | -0.5 |
| S25-3 | 0.08 | 0.01 | 0.15 | 0.04 | 0.5 | 0.4 | -40 | -0.07 | -45 | -0.2 | -90 | -7.5 |

(within $\pm\,0.3\,\mathrm{cm}^{-3}$) with observations in 4 out of 5 simulations. Only the S25-2 simulation strongly underestimates the
maximum ice crystal number concentration by $1\,\mathrm{cm}^{-3}$ (see Sect. 3.2.1 and Sect. 3.2.2), whereas the simulated median ice crystal number concentrations match the observations well. This is not the case for the other four simulations, where the median concentration is underestimated by an order of magnitude. When we also consider the mean values, we see that the model in general has only a few grid cells with high ice crystal number concentrations, while a lot of grid cells have very low ice crystal number concentrations. Regarding the changes in cloud droplets, the model fails to reproduce the maximum
cloud droplet reductions, where 4 out of 5 simulations show almost no reduction. Only in the simulation S26-2.5a a stronger reduction in cloud droplet number concentration is notable. However, for all simulations the median and mean cloud droplet reductions are strongly underestimated.

As a next step, we aim to identify when suitable conditions for the WBF process exist inside the seeding plume by applying a set of theoretical equations (Korolev and Mazin, 2003; Korolev, 2007):

$$w' < w < w^* \text{ with } w' = \frac{e_{s,i} - e_{s,w}}{e_{s,w}} N_w \overline{r}_w \chi \text{ and } w^* = \frac{e_{s,w} - e_{s,i}}{e_{s,i}} N_i \overline{r}_i \eta \qquad (2)$$

where $w$ is the vertical velocity, $w^*$ and $w'$ are the computed vertical velocity thresholds, $e_{s,w/i}$ are the saturation vapor pressures with respect to water and ice, respectively, $N_{w/i}$ is the number concentrations for cloud droplets and ice crystals, respectively, and $\overline{r}_{w/i}$ are the mean radius for cloud droplets and ice crystals, respectively. $\eta$ and $\chi$ are terms dependent on ambient temperature and pressure used to calculate the thermodynamic equilibrium. For $w > w^*$, both cloud droplets and ice
crystals grow, while for $w < w'$ they shrink.

Conditions favorable for the WBF process are therefore constrained by the vertical velocity and the integral ice crystal and cloud droplet radius. For positive vertical velocities, i.e. updrafts, the ice crystal number concentrations and mean radii define the WBF regime, while for negative vertical velocities, i.e. downdrafts, cloud droplet number concentrations and mean radii

define the WBF regime. Hence, for $0 < w < w^*$ the ice phase acts as a sink for the supersaturation generated by the updrafts. For $0 > w > w'$ the liquid phase is constraining the WBF process as the downdrafts reduce the supersaturation. As long as cloud droplets are present, they serve as a source for the ice crystals to grow. We call these two conditions updraft WBF (WBF$_\uparrow$) and downdraft WBF (WBF$_\downarrow$). To quantify the occurrence of the WBF process, we determined for each model grid box inside the seeding plume which of these two regimes prevails (Fig. 10a-g). For most of the shown timesteps, roughly 50 % or more of the plume grid boxes are within the WBF regime. About 40 % of the time both cloud droplets and ice crystals can grow inside the plume ($w > w^*$). The simultaneous evaporation and evaporation ($w < w'$) of the hydrometeors occurs least often (<2 % of the time) within the seeding plume. During WBF conditions the WBF$_\downarrow$ is dominant, which is further supported by the vertical velocities in Fig. 10h-n, which indicate the presence of downdrafts inside the seeding plume. We note here, that we cannot distinguish between the microphysical (latent heat release) and dynamical (topography and wind field) influence on ice crystal growth and evaporation of cloud droplets. Henneberger et al. (2023) discussed that some updraft invigoration may occur due to latent heat release upon ice nucleation, however this is still under debate.

Based on the prognostic mass and number mixing ratio for the ice and cloud droplet phase, we calculated the ice crystal and cloud droplet radii assuming a spherical shape for all particles (Fig. 10o-ab). We can see that the mean ice crystal radius increases over time, while the mean cloud droplet radius slightly decreases, indicating that in fact the WBF process takes place. Also the mean radius of cloud droplets in the reference simulation is consistently larger than in the seeding simulation (see Fig. D1). Additionally, we show the mean radius for ice crystals and cloud droplets measured by HOLIMO (see Sect. 2.1) during the seeding event (see $t_2$). While the the cloud droplet radius distributions match well (Fig. 10w), the model underestimates the mean ice crystal radius by almost 30 μm (Fig. 10p). Also, the observations show a broader range of ice crystal radii than was simulated. This clearly indicates that the WBF process in the model is too slow to reproduce the observed growth rates in the field and/or the turbulent motions are too weak. Finally, we diagnosed the rate of water vapor deposition onto the ice crystals during the model simulation (Fig. 10ac). Initially, we simulated the largest vapor deposition growth rates (up to $1.5 \times 10^{-4}\,\mathrm{g\,m^{-3}s^{-1}}$) with ice crystals growing the fastest, which can also be seen in the evolution of the ice crystal radius over time (Fig. 10o-ab). At later model output time steps, the rate decreases to almost $0\,\mathrm{g\,m^{-3}s^{-1}}$, when ice crystal no longer grow due to vapor deposition.

### 3.3.1 Sensitivity analysis of seeding particle emission rate

Here we perform a sensitivity analysis to investigate the effect of the seeding particle concentration on the ice crystal number concentrations and cloud droplet reductions (Fig. 11). With $10^6\,\mathrm{m^{-3}s^{-1}}$ seeding particles, the model predicts the observed ice crystal number concentrations, however, it fails to match the observed reductions in cloud droplets (Sect. 3.3, Fig. 11a). When increasing the seeding particle emission rate by a factor of three, we see that the model overestimates the ice crystal number concentrations while still underestimating the cloud droplet reductions (Fig. 11b). Only when we introduce 10 times more seeding particles than in the default configuration, the model reaches the observed cloud droplet reductions but yields much higher ice crystal number concentrations (Fig. 11c). This further supports the hypothesis that the model does not represent ice crystal growth rates accurately. Only if a high number of ice crystals is present, the observed cloud droplet reductions can

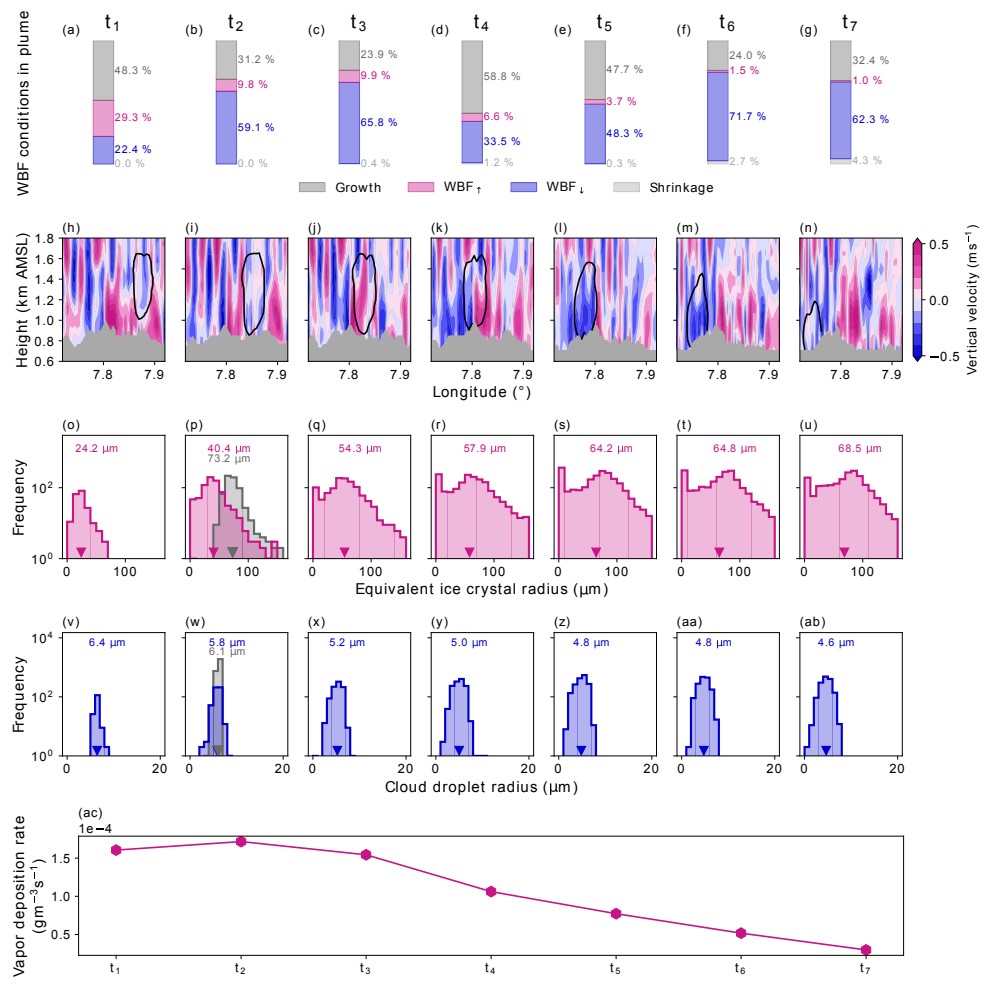

**Figure 10.** First row (**(a)-(g)**): Percentages of WBF and non-WBF conditions inside the seeding plume for experiment S26-2.5a for different model output time steps ($t_1$ - $t_7$). Conditions were determined based on Korolev and Mazin (2003) and Korolev (2007) and differentiated between "Growth" (dark grey, both cloud droplets and ice crystal grow), "WBF$_\uparrow$" (pink, WBF defined by updrafts), "WBF$_\downarrow$" (blue, WBF defined by downdrafts), and "Shrinkage" (light grey, cloud droplets evaporate and ice crystals sublimate). Second row (**(h)-(n)**): Cross sections of vertical velocity (instant values) along the mean wind direction over time. The black contours indicate the location of the seeding plume and the grey shading denotes the topography. Third row (**(o)-(u)**): Frequency distributions of equivalent ice crystal radius (μm, pink) over time and mean equivalent radius (downward facing triangle, pink numerical value) for the seeding plume at every model output time step. Panel **(p)**, in addition, shows the mean equivalent ice crystal radius distribution (grey) measured by HOLIMO for the time period in Fig. 9b. Fourth row (**(v)-(ab)**): As in the third row but for cloud droplet radius (μm). Panel **(w)** depicts also the measured cloud droplet radius distribution (grey) measred by HOLIMO for the time period in Fig. 9b. Last row (**(ac)**): Simulated rate of water vapor deposition onto ice particles ($gm^{-3}s^{-1}$) over time. $t_2$, $t_4$, and $t_6$ correspond to the model responses shown in Fig. 9c-e.

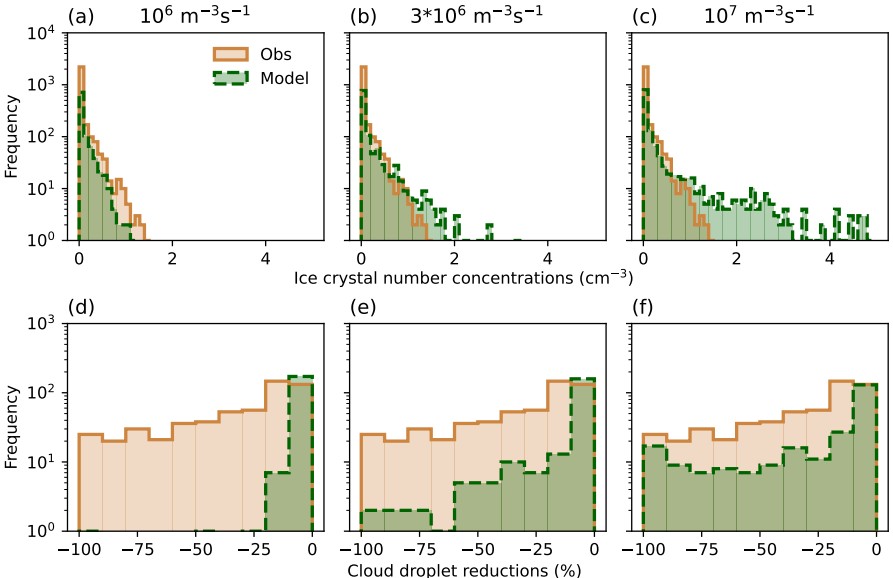

**Figure 11.** Upper row: Ice crystal number concentrations $(\mathrm{cm}^{-3})$ of experiment S26-2.5a observed (brown, solid) by HOLIMO and simulated (green, dashed) with $10^6\,\mathrm{m}^{-3}\mathrm{s}^{-1}$ seeding particles (default seeding particle emission rate, **(a)**), $3*10^6\,\mathrm{m}^{-3}\mathrm{s}^{-1}$ seeding particles (**(b)**), and $10^7\,\mathrm{m}^{-3}\mathrm{s}^{-1}$ seeding particles (**(c)**). The bin size is $0.1\,\mathrm{cm}^{-3}$. Lower row: Relative changes in cloud droplet number concentrations (%) observed (brown, solid) and simulated (green. dashed) for the corresponding seeding particle concentrations. The bin size is $10\,\%$. Note that for all panels, the observations are identical. For the model response, the difference between a seeding and a reference simulation was taken at 10:30 UTC (as in Fig. 9c).

be reproduced. Also the high ice crystal number concentrations are only present in a few grid cells (see mean and median in Table 3). Even with the highest seeding particle emission rate, the median ice crystal number concentration still stays below
the observed median. However, the mean is higher than the observed one pointing towards a shift in the spatial distribution of ice crystals in the plume compared to the other simulations (see Table 2).

Based on the results from this sensitivity analysis regarding the seeding particle emissions rate, we see that an emission rate of $10^6\,\mathrm{m}^{-3}\mathrm{s}^{-1}$ is a good approximation to conduct seeding experiments in the model and to compare them to our observations in the field. We also tested lower seeding particle concentrations, but then the ice crystal number concentrations
were underestimated. The seeding particle emission rate of $10^6\,\mathrm{m}^{-3}\mathrm{s}^{-1}$ may still be an overestimation in seeding particles due to the warmer temperatures in the simulated boundary layer clouds (see Fig. 3), which leads to lower ice nucleation rates of the seeding particles given the strong temperature dependence of ice nucleation. Hence, if the model were colder, we would simulate a higher ice crystal number concentrations. We are aware of this limitation but decided to constrain our seeding setup by the observed ice crystal number concentrations instead of the seeding particle emission rate, as the latter cannot be precisely
estimated.

**Table 3.** Median, mean, and maximum observed (Obs) and predicted (Model) ice crystal number concentrations ($cm^{-3}$) and cloud droplet reductions (%) for the simulation S26-2.5a and the two simulations with increased seeding particle emissions rates ($3*10^6\,m^{-3}s^{-1}$ and $10^7\,m^{-3}s^{-1}$, respectively). For the observations we considered the entire measurement period during the seeding experiment. For the model we considered the output time step that is closest to the time of the expected seeding signal at the field site, i.e. based on the calculated growth time in Table 1. Note that the observations are identical.

| | Ice crystal number concentration ($cm^{-3}$) | | | | | | Cloud droplet reduction (%) | | | | | |
| | Median | | Mean | | Maximum | | Median | | Mean | | Maximum | |
| Name | Obs | Model | Obs | Model | Obs | Model | Obs | Model | Obs | Model | Obs | Model |
| --- | --- | --- | --- | --- | --- | --- | --- | --- | --- | --- | --- | --- |
| S26-2.5a ($1*10^6$) | 0.2 | 0.02 | 0.27 | 0.10 | 1.0 | 1.2 | -20 | -0.5 | -30 | -2.00 | -100 | -90 |
| S26-2.5a ($3*10^6$) | 0.2 | 0.03 | 0.27 | 0.20 | 1.0 | 3.3 | -20 | -1.0 | -30 | -7.00 | -100 | -90 |
| S26-2.5a ($1*10^7$) | 0.2 | 0.05 | 0.27 | 0.45 | 1.0 | 10.0 | -20 | -1.2 | -30 | -15.00 | -100 | -90 |

## 4 Conclusions

This study presented LES in the scope of the CLOUDLAB project aimed at reproducing the field seeding experiments and constraining the WBF process inside the model. For that, a nested LES was set up to conduct seeding simulations with the numerical weather model ICON, which includes an implementation of a new seeding parameterization for freezing of AgI
particles. Five seeding experiments from two days were simulated using a single-day simulation as a surrogate, because the model failed to reproduce low enough temperatures in the boundary layer. The experiments differed in seeding temperature and distance, allowing us to investigate the temperature sensitivity of the seeding parameterization and the effect of different seeding distances from the field site on ice crystal growth and dilution.

We first showed that ICON is able to reproduce long-lasting low-level clouds but with a weaker temperature inversion. The
420 observed seeding temperature was nevertheless simulated, enabling us to conduct seeding simulations in the model. The first two simulations were conducted at a seeding temperature of -6.5 °C (S26-2.5a, S26-2.5b) and are in good agreement with observed ice crystal number concentrations obtained from the in situ device HOLIMO. In addition, the simulated extent of the seeding plume agrees qualitatively with the observations from radar scans. The measurements for the three experiments at warmer temperatures (-5.5 °C) with varying distances from the field site (2, 2.5, and 3 km) show a strong dilution effect
with maximum ice crystal number concentrations up to $2.5\,cm^{-3}$ for the shortest seeding distance down to $0.5\,cm^{-3}$ for the furthest seeding distance. The model strongly underpredicts the ice crystal number concentrations for one of the simulations which may originate either from the fact that the freezing parameterization does not adequately represent the frozen fractions at warmer temperatures, or that the ice nucleating activity was limited by available cloud droplets.

In the second part, we investigated the WBF process at high spatio-temporal resolution. The in situ measurements showed
that the high ice crystal number concentrations were accompanied by reductions in cloud droplet number concentrations confirming that the WBF process took place. During the seeding experiments, we observed fully glaciated patches, i.e. where zero

cloud droplet number concentrations with high ice crystal number concentrations were measured. By analyzing the relative cloud droplet reductions within the seeding plume with regard to the undisturbed background, we computed frequency distributions of observed reductions in cloud droplets. This was compared to the simulated cloud droplet number concentrations by taking the difference between a seeding simulation and a reference simulation without seeding. We showed that the model could not reproduce the observed strong reductions in cloud droplet number concentrations, supporting the hypothesis that the WBF process is too slow in the model or that tubrulent motions, which could locally enhance growth rates, are too weak. Only at later model output time steps, comparable cloud droplet reductions were achieved. By calculating the proportion of favorable conditions for the WBF process to take place inside the seeding plume and identifying the changes in cloud droplet and ice crystal radii, we showed that the WBF process takes place in the model but at a slower rate than observed in the field. We also tested the effect of increased seeding particle emission rate on ice crystal and cloud droplet changes. Tripling the seeding particle emission rate led to a slight overestimation in ice crystal number concentrations, while the reduction in cloud droplets still could not be reproduced. Only by introducing 10 times more seeding particles, we reached comparable cloud droplet reductions as in the observations. However, in this case, the ice crystal number concentrations were largely overestimated. This further supports the hypothesis that the WBF process in the model is too slow or its turbulent enhancement too weak, which in turn has implications on the efficiency of precipitation formation.

Further work will build upon these findings by perturbing the parameterization for ice crystal growth through vapor deposition via the ventilation coefficient to account for turbulent motions and by quantifying the impact of turbulence on ice crystal growth. With the final field campaign, we expect to extend our current experimental data set to have more variation in environmental conditions to further constrain ice formation and growth. This study shows the high potential of the gathered CLOUDLAB data in conjunction with a modeling approach to better understand ice crystal growth.

*Code and data availability.* Analysis and plotting scripts are archived at https://doi.org/10.5281/zenodo.10990787 (Omanovic et al., 2024a). Generated data are archived at https://doi.org/10.5281/zenodo.10991224 (Omanovic et al., 2024b). At the time of the analysis and writing of this manuscript, the ICON code was only available upon request for anyone with an ICON license. In the meantime, an open source version of ICON was released (https://www.icon-model.org), which can be used to perform the simulations.

## Appendix A: Seeding plume tracking

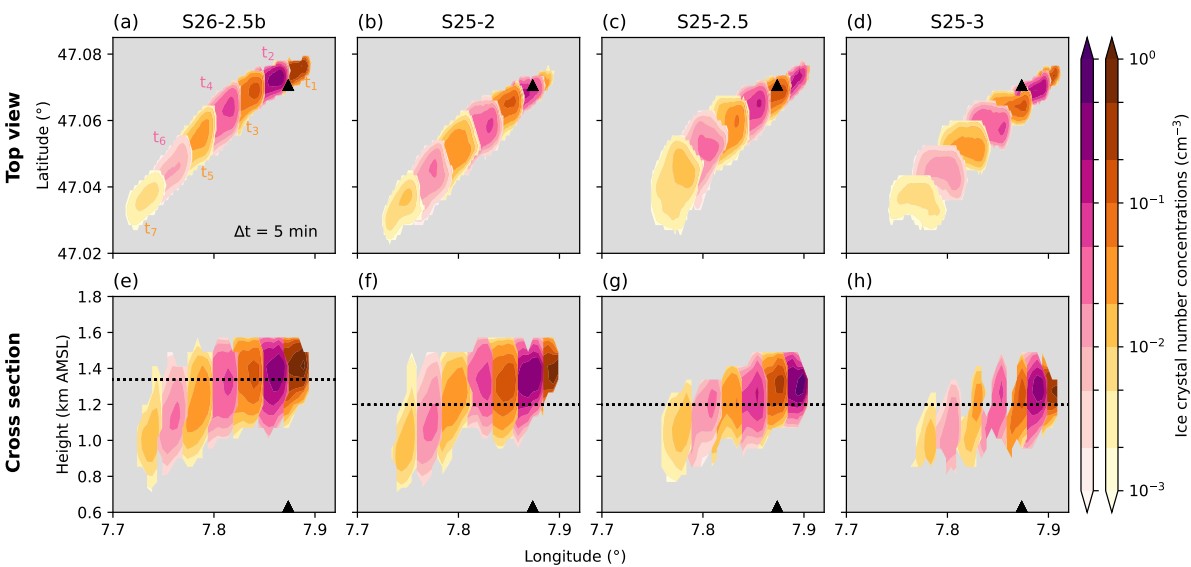

**Figure A1.** Simulated ice crystal number concentration changes (cm$^{-3}$, colormap) after seeding in simulations S26-2.5b (first column), S25-2 (second column), S25-2.5 (third column) and S25-3 (fourth column) with **(a), (b), (c), (d)** showing the top view at seeding height (dashed line in **(e), (f), (g), (h)**): $t_1$ is the first five-min output time step after seeding start. Alternating colors are used for better visibility between the output time steps. **(e), (f), (g), (h)** show the vertical cross sections in ice crystal number concentrations (cm$^{-3}$) along the mean wind direction at the seeding height for each output time step. The black triangle denotes the field site in all panels.

## Appendix B:  Cloud droplet number concentration comparison

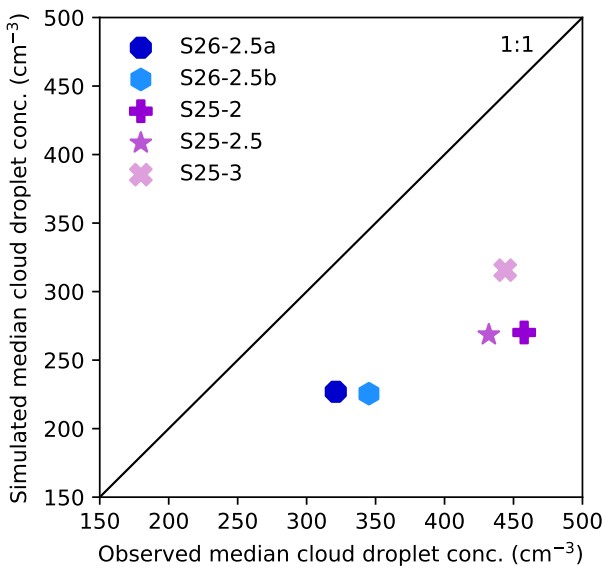

**Figure B1.** Comparison of simulated (reference) (y-axis) and observed (x-axis) median cloud droplet number concentration ($cm^{-3}$) for all seeding simulations. The median was taken over 20 min of unperturbed model simulation and HOLIMO observations.

## Appendix C: WBF investigation

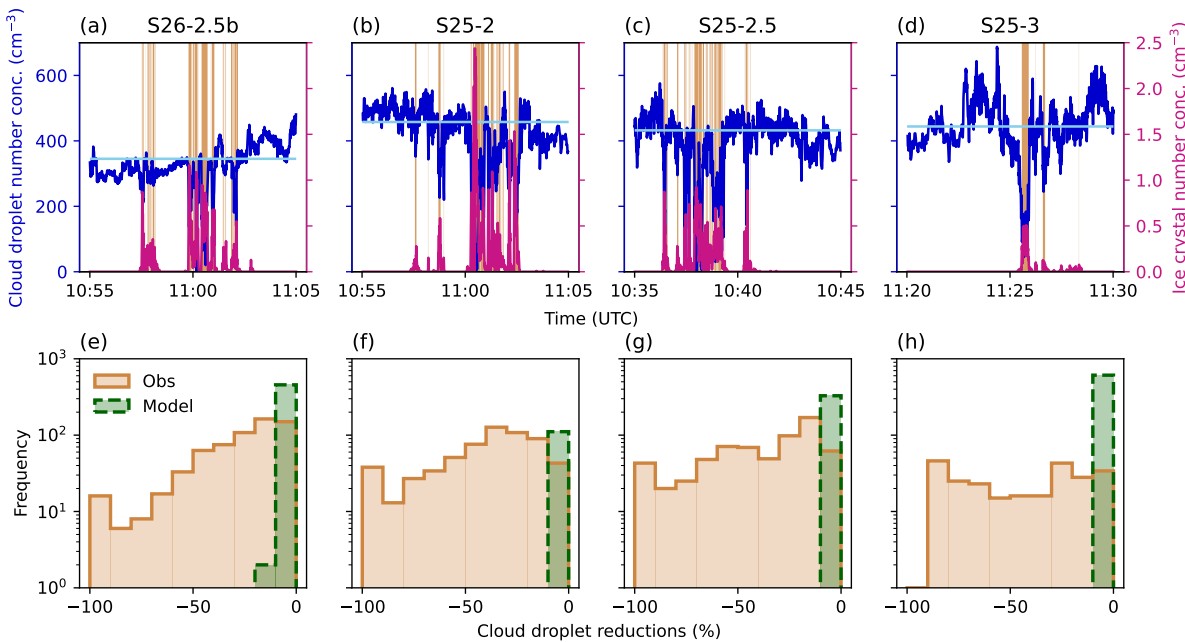

**Figure C1.** Upper row: Cloud droplet ($\mathrm{cm}^{-3}$, blue) and ice crystal ($\mathrm{cm}^{-3}$, pink) concentrations measured by HOLIMO during the four experiments S26-2.5b (**(a)**), S25-2 (**(b)**), S25-2.5 (**(c)**), and S25-3 (**(d)**). Horizontal light blue lines show for each experiment the median cloud droplet number concentration over 20 min time span of the background cloud prior/after seeding. The brown shading indicates time periods where the ice crystal number concentration was above $0.001\,\mathrm{cm}^{-3}$ and which were considered for the cloud droplet reduction analysis. The frequency distribution for the observed (brown, solid) and simulated (green, dashed) cloud droplet reductions are shown in the lower row (**(e)-(h)**). The bin size is set to $10\,\%$.

## Appendix D: Cloud droplet radius from reference simulation

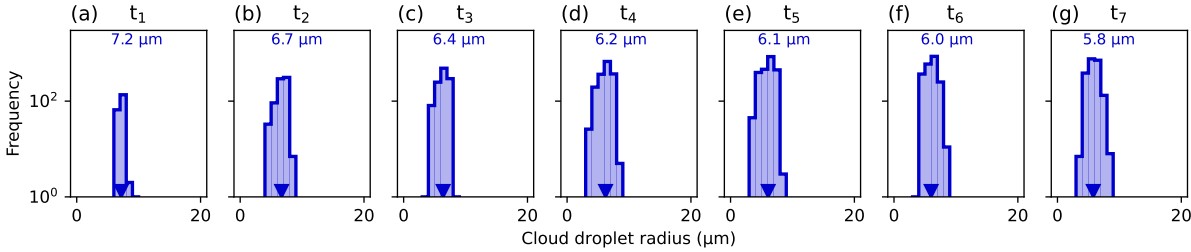

**Figure D1.** Frequency distributions of cloud droplet radius (μm, blue) over time and mean radius (downward facing triangle, blue numerical value) for the seeding plume at every model output time step for the reference simulation of S26-2.5a (no seeding).

*Author contributions.* UL, JH, FR conceived of the idea of CLOUDLAB and obtained funding. NO, CF, JH, AJM, FR, RS, HZ designed and conducted the in-cloud seeding experiment presented here, with conceptual input from UL. CF conducted the entire analysis of HOLIMO data. KO, PS provided the vertically pointing radar used for model verification. NO, SF implemented the AgI freezing parameterization in ICON. NO set up the model nesting, performed all simulations and subsequent analysis with the here presented figures. NO, SF, UL contributed to the interpretation of the results. NO wrote the manuscript. All authors contributed to the editing and review of the manuscript.

*Competing interests.* The authors declare that they have no conflict of interest.

*Acknowledgements.* The CLOUDLAB project has received funding from the European Research Council (ERC) 411 under the European Union's Horizon 2020 research and innovation program (grant agreement 412 No. 101021272 CLOUDLAB). This work was supported by a grant from the Swiss National Supercomputing Centre (CSCS) under project ID s1144. We further want to thank Stephanie Westerhuis and Brigitta Goger for their support during the model setup by providing model grids and scripts to generate the initial and boundary conditions and their continuous helpful input regarding model validation.

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
