# Peer review of "Evaluating the Wegener-Bergeron-Findeisen process in ICON in large-eddy mode with in situ observations from the CLOUDLAB project"

_EGUsphere, 2023_

## Author Comment (AC1)

**Authors' comments to Anonymous Referee #1**

We thank the Anonymous Referee #1 for their valuable feedback on our manuscript, which improved the quality of the manuscript significantly, and we address the raised points below in blue.
* * *
The authors conducted large-eddy simulations to explore the impact of seeding on boundary layer supercooled clouds. The model setup is based on observations in the CLOUDLAB project. They first demonstrated the capability of the model to simulate and reproduce the seeding experiments at different environmental conditions. Then, they investigated the WBF process in the model by changing the INP emission rate. One conclusion is that the WBF process seems to be less efficient in the model than in the field. The conclusion is striking and interesting. One inconsistency is that the seeded cloud is expected to be above the site at 10:30 UTC (see Fig.6). However, in-situ measurements show that ice particles exist at around 10:35 UTC (see Fig. 9). So the apparent less efficient WBF process in the model might be due to some other reasons, e.g., underestimation of the growth time or advection time. See explanation to Line 149.

In general, the manuscript is well written and easy to read. I have some minor comments listed below.

Line 113: "both experiments are identical in their setup". Since both seeding experiments are at the same location, is there any physical reason why there are two seeding experiments on that day? For example, I can understand S25-2, S25-2.5, S25-3 can test the impact of distance (growth time), but what about S26-2.5a and S26-2.5b? What can we learn from these two experiments?
In the field we conducted experiments using an identical setup to test the validity of the signal we observe in the radar and in-situ observations (similar to doing exact replicates in laboratory experiments). Here, we use both again to show that the model can reproduce the seeding signal consistently in the two experiments, but also when changing the seeding distance (other three experiments).

Line 149: "The frequency of model output was set to 5 min". The ice growth time is between 6 and 9 min (Table 1). Please comment on whether the relatively low output frequency would affect the comparison between observation and simulation.

In the table below we show the times of the expected arrival time and the closest model time step. The plume arrives at the field site almost always at a 5 min time step. We also tested 1 min output frequency for the simulation S26-2.5 and the results are similar. Hence, we follow here the 5 min output frequency. With the expected arrival time at the field site we also chose our model time step which may disagree with the time period of the observations given the differences / uncertainties in wind speed.

| Name | Seeding start (UTC) | Growth time (min) | Arrival at field site (UTC) | Closest model time step (UTC) |
|---|---|---|---|---|
| S26-2.5a | 10:22 | 8.0 | 10:30 | 10:30 |
| S26-2.5b | 10:48 | 7.1 | 10:55 | 10:55 |
| S25-2 | 10:50 | 6.1 | 10:56 | 10:55 |
| S25-2.5 | 10:28 | 8.0 | 10:36 | 10:35 |
| S25-3 | 11:15 | 9.1 | 11:24 | 11:25 |

Table 1: Please also add the seeding height in the table. It is difficult to accurately read the seeding height from Figure 4.

We've added the seeding heights from the field and the model to Table 1.

Line 178: "seeding particle emission rate". Please add more justification of the choice of emission rate. For example, is it based on the estimation of the real seeding experiments, or is it chosen to match the ice number concentration. I find some discussions about it in the later part of the manuscript, but it is better to add some justifications here.

We added the following at the end of the paragraph (Line 201): *"The seeding particle emission rate and thus seeding setup were constrained by the ice crystal number concentrations observed by HOLIMO and tuned in such a way that they match for the seeding simulation S26-2.5a (Sect. 3.3.1)."*

Line 235: "There is a good qualitative agreement between …" What is the scanning frequency of the radar? How does reflectivity from the scanning radar look like e.g., 5 min before and 5 min after 10:30 UTC? Can the radar observation show the impact of cloud seeding?

In Henneberger, Ramelli et al., (2023), we show that the seeding signal can be observed by a vertically pointing radar (their Fig. 6) for several minutes and in sector scans (their Fig. 9). The seeding signal is clearly distinguishable from the background due to increased reflectivity values (background: -25 dBZ, seeding: -10 dBZ). The scanning frequency of the radar shown in the manuscript is 90 s with a scan speed of 1° per second allowing for several scans of a single seeding plume including parts of the background as well. We added the scanning frequency to the figure caption: "**Figure 6.** Comparison of the radar reflectivity measured by a scanning cloud radar with a scanning frequency of 90 s per scan (Mira-35, Metek, (a)) …".

Line 296, 345 "(not shown)" is not accepted nowadays. Please consider adding the figure in the supplementary material or rephrase the sentence.

We added the figure for the cloud droplet number concentration comparison to the supplement (Fig. B1) and removed the second "(not shown)" in Sect. 3.3.1 as it is not needed.

Line 296. "The ice crystal number concentrations are in good agreement (within +-0.3 cm-3) with observations in 4 out of 5 simulations." What I see is that the simulated median ICNC is one order of magnitude smaller than the observation, while the maximum value is similar. Even if the median ICNC is 0 from the model, the uncertainty is still within 0.3 cm-3. So I think this statement is not accurate.

We adapted the text as follows (Line 326): "*The maximum ice crystal number concentrations are in good agreement (within ± 0.3 cm$^{-3}$) with observations in 4 out of 5 simulations. Only the S25-2 simulation strongly underestimates the maximum ice crystal number concentration by 1 cm$^{-3}$ (see Sect. 3.2.1 and Sect. 3.2.2), whereas its median ice crystal number concentrations match well with the observations. This is not the case for the other four simulations, where the median concentration is underestimated by an order of magnitude. When we also consider the mean values, we see that the model in general has only a few grid cells showing the high ice crystal number concentrations, while a lot of grid cells have very low ice crystal number concentrations. Even though the seeding plume spreads out over several levels (see Fig. 5), the internal mixing inside the plume seems to be inefficient leading to this discrepancy. Regarding the changes in cloud droplets, the model fails to reproduce the maximum cloud droplet reductions, where 4 out of 5 simulations show almost no reduction. Only in the simulation S26-2.5a a stronger reduction is notable. However, for all simulations the median and mean cloud droplet reductions are strongly underestimated.*"

---

## Author Comment (AC2)

**Authors' comments to Anonymous Referee #2**

We thank the Anonymous Referee #2 for their valuable feedback on our manuscript, which improved the quality of the manuscript significantly, and we address the raised points below in blue.
* * *
OVERALL COMMENTS:

Omanovic et al. evaluated the WBF process rate in their model by performing ice seeding simulation and comparing the simulation results with the observations. A few major issues may affect the value of this paper and need to be addressed before I can recommend this paper for publication:

- Assuming the main conclusion of this paper (that is, the WBF process rate in the model is much slower than those inferred by comparison with observation) is correct, is it merely a manifestation of the well-known fact that spherical ice assumption underestimates growth rate by vapor deposition?
See answer to specific comment (L137)

- While the main conclusion is not surprising, the method leading to it is problematic: Given the uncertainties in simulated meteorology and microphysics and observations, it is concerning that the authors essentially attributed all discrepancies between the simulations and the observations (regarding cloud droplet depletion) to WBF. For example, the authors dismissed several discrepancies between the simulations and the observations as unimportant without sufficient evidence; they used Jan. 26 simulation as a testbed for seeding and compared that with observations from Jan. 25 without enough justification; they tuned the seeding rate to match observed ice number concentration, inviting compensating errors.
See answer to specific comment (L194)

- The evidence and reasoning supporting the main conclusion are not very clear. See detailed comments below.

DETAILED COMMENTS:

L26: During WBF, it is the evaporation of liquid that maintains supersaturation w.r.t. ice.
L27: How is the ice crystal radius defined?
L28: What does "which" refer to?
We addressed the above three points as follows (Line 27): "*Whether $e$ exceeds $e_{s,w}$ or not depends, among other factors, on the vertical velocity as a source for water vapor, and on the integrated ice crystal surface (ice crystal number concentration × mean ice crystal radius), where the present ice crystals deplete the supersaturation by consuming the available water vapor generated by evaporating cloud droplets (Korolev and Mazin, 2003; Korolev, 2007). Here, Korolev and Mazin (2003) define the ice crystal radius as the half of the maximum dimension of the particle.*"

L65: Why "on the statistical significance"?
Given the natural variability in cloud systems, it is difficult to assess the feasibility of cloud seeding in an experimental setup. Here, numerical weather models can help to assess the uncertainty of cloud seeding by conducting repeated seeding simulations.

L74: This is confusing. Did the cited works use a weather model or LES?
The model used in these studies is the WRF model, which can be set up in various horizontal resolutions ranging from several kilometers down to several meters. With 100 m resolution they are in the range of large-eddy simulations, similar to this study. We adapted the following sentence to make it clear it is a weather model used in high-resolution setup, and not an idealized LES (Line 77): *"They employed a weather model in 100 m horizontal resolution, i.e. they conducted non-idealized large-eddy simulations (LES), and reproduced the environmental conditions and the dispersion of the seeding plume (Xue et al., 2016; Chu et al., 2017; Xue et al., 2022)"*.

L85: "CLOUDLAB aims to ...": This is unclear. How to improve precipitation forecast skill by evaluating seeding method?
We have adapted the sentence as follows to highlight that we focus on ice crystal growth inside the cloud by perturbing supercooled clouds (Line 89): *"By improving the parameterizations of ice crystal growth in ICON with updated equations, CLOUDLAB aims to increase precipitation forecast skills of numerical weather prediction models by first evaluating the ice crystal growth rate in seeded supercooled clouds in a high-resolution model"*.

L107: Sorry if I missed this: is the seeding flare mounted on the UAV?
We adapted the sentence in the introduction as follows (Line 84): *"To conduct the seeding experiments, an uncrewed aerial vehicle (UAV) attached with a seeding flare is flown into the cloud to release seeding particles (containing AgI) upstream of the field site.".*

L135: "dynamically almost stable and persistent low stratus clouds": Did the authors mean the clouds reside in a stable layer?
Yes, the cloud is capped by a strong temperature inversion (see Fig. 3, the radiosonde profiles) and is characterized by north-easterly winds. This leads to the persistency of low stratus over the Swiss Plateau, and with that to the assumption of almost dynamically stable conditions (compared to other clouds). We adapted the sentence as follows (Line 139): *"… due to persistent low stratus clouds in a stable boundary layer."*

L137: Sections 2.2 and 2.3: Please describe the ice properties used in the model/parameterization, including size ~ fall velocity relationship, density, size distribution, ice shape, and so on.
We added a short description of the two-moment microphysics to Sect. 2 as follows (Line 156): *"In the following we provide a short description of the two-moment microphysics scheme used in the model. For more details, please refer to Seifert and Beheng (2006). The scheme tracks mass and number mixing ratios for six hydrometeors: cloud droplets, ice crystals, snowflakes, graupel, hail, and rain drops by assuming a Gamma-distribution for the underlying size distributions. All relevant cloud processes are parameterized, such as cloud droplet activation, ice crystal nucleation, growth of ice particles by water vapor deposition*

*(i.e., the WBF process), riming, melting, and sublimation of ice crystals. The maximum diameter and terminal fall velocity are parameterized following power laws with constant coefficients (Seifert and Beheng, 2006). As we are investigating the ice processes within mixed-phase clouds, we provide additional information relevant to ice particles. The ice crystal shape is set to be spherical, which is a simplification in the scheme given the large variety of shapes (Bailey and Hallett, 2009). In this study, we do not change the shape of the ice crystals as we want to investigate the ice crystal growth rate in the default configuration of the model. During the conducted seeding experiments, we measured mostly needles or columns. When we compare the ice crystal sizes in Fig. 10, we investigate the mean equivalent radius of ice crystals.".*

L140: "80 vertical levels": What is the vertical grid spacing around the cloud layer?
We added the following information (Line 146): *"Both inner nests also have 80 vertical levels and a vertical grid spacing ranging from approximately 20 m (at cloud base) to 80 m (at cloud top).".*

L149: Schmale et al. (2018) reported CCN concentrations at many sites with different characteristics and seasonality. Which site and season did the authors use as a reference?
We used the stations Melpitz (Germany, continental background), CESAR tower (the Netherlands, near coast, rural background), and Vavihill (Sweden, rural background) as a reference for the winter months December, January, and February. Our field side is located in rural Switzerland, and these stations seemed to be a good proxy. We adapted the sentence as follows (Line 153): *"The cloud condensation nuclei concentration was set to 1000 cm$^{-3}$ following Schmale et al. (2018) for rural and continental areas (Melpitz, Germany; CESAR tower, the Netherlands; and Vavihill, Sweden) during wintertime and is uniformly distributed in the domain.".*

L151: "highlight the impact of growth time on the seeding plume": This is confusing; please clarify.
We adapted the sentence as follows (Line 168): *"The simulations S25-2, S25-2.5, and S25-3 highlight the impact of the seeding distance on the ice crystal growth time.".*

L156: "utilize the simulation of the 26 January 2023 for both seeding days": This is unclear; please clarify.
Given the discrepancy in temperature between model and observations on 25 January, we decided to conduct the seeding simulations of the 25 January during the simulation of 26 January, because here the simulated temperatures agreed well with observations. We adapted the sentence as follows (Line 173): *"For this reason, we decided to utilize the simulation of 26 January 2023 for all seeding experiments conducted on 25 and 26 January 2023 (see Sect. 3.1 for detailed discussion).".*

L168: "given the similar response in freezing for the larger particle sizes": But 400 nm is an upper bound for both Henneberger's field experiment and Marcolli's lab measurements. What about smaller particles?
Marcolli et al. (2016) show similar freezing curves for 40 nm, 50 nm, 200 nm and 400 nm sized particles. Henneberger, Ramelli et al. (2023) found that the mean particle diameter is between 100 and 400 nm. This is why, we follow the freezing curve for 400 nm. There are

smaller particles, but as Marcolli et al. (2016) shows their freezing ability is smaller, they may be less relevant for our seeding temperatures. We adapted the text as follows (Line 185): *"However, for sizes larger than 40 nm the freezing curves are fairly similar, thus we decided to follow the 400 nm measurements."*

L170: Does the probability of freezing depend on time?
No, it is a deterministic approach, such that a fraction of the available aerosol particles will freeze when the conditions (below a given temperature and being inside a cloud) are met. We added the following to the methods description Sect. 2.3 (Line 179): *"To simulate glaciogenic cloud seeding in ICON, we implemented a deterministic freezing parameterization specifically for the seeding particles (AgI) used in the field."*

L187: "The seeding plume was defined ..." If the seeding plume is defined by a threshold, why did the authors still need the unseeded simulation as the background?
We need an unseeded simulation (reference simulation) to identify the changes in, e.g., cloud droplet number concentrations and vertical velocities for the grid cells affected by the seeding and not elsewhere in the model domain.

L191: Again, do AgI particles freeze immediately after the release? If not, the time between the release and the arrival is not the growth time.
We assume immediate nucleation of the seeding particles given the measurements by Marcolli et al. (2016) showing high ice nucleation activity below -5 °C and the high hygroscopicity of the particles (Henneberger, Ramelli et al., 2023). We added an additional sentence in Sect. 2.3 (Line 192): *"For the analysis, we assume immediate ice nucleation given the high ice nucleation activity of AgI below -5 °C."*.

L194: There are a few issues in Section 3.1. First, both Figures 3 and 4 showed that the temperature range where the seeding occurred was suitable for secondary ice production. Is this process parameterized in the model? Does it have an impact on the results? Second, the use of Jan. 26 as a testbed for Jan. 25 and then comparing the simulation results with Jan. 25 observations is questionable. Even though the seeding height is adjusted to match what actually occurred on Jan. 26, the meteorology for Jan. 26 seems to be different from Jan. 25 (cloud temperature range, thermodynamic profiles, maybe also liquid water content profile, etc.) Please carefully justify this decision.
Secondary ice production is simulated in the model following Hallet and Mossop. However, it only occurs if graupel or hail particles are rimed, and then a splintering rate is calculated. In our simulations the amount of graupel particles is close to 0, so we do not see an effect of SIP.
Given that the model simulation from 26 January 2023 reproduces the cold temperatures from 25 January 2023, without the sharp inversion at cloud top, we believe it is adequate to use the 26 January 2023 simulation as a testbed. While we do have a lower cloud on 25 January 2023, we still encounter a persistent low stratus cloud with north-easterly to easterly winds as shown in Table 1. The wind speeds are comparable in the observations and the model, and this further supports our method to use 26 January 2023 as a surrogate model.

L209: "predicted cloud cover": How did the authors define cloud cover from radar observations and simulations?
The reflectivity of the cloud radar (FMCW-94-DP, Radiometer Physics GmbH) is used as a proxy for cloud cover observed at the field site. In the model, the cloud cover is diagnosed based on the prognostic cloud water mass.

L210: "a long-lasting low cloud that reaches similar cloud top heights as observed": But previously the authors said that the lower inversion base led to a lower cloud top.
We have adapted the text as follows (Line 234): *"We see that the model predicts a long-lasting low cloud that reaches slightly lower cloud top heights than observed by the radar with the seeding simulations still being fully inside the cloud. The lower cloud top can be also seen in the comparison of relative humidity in the radiosonde profiles."*.

L210: I probably missed it, but would you please described seeding height in numbers in the text or a table or as annotation in Figure 4 in addition to showing triangles in Figure 4?
We added the seeding heights from the field data and the model simulations to Table 1.

L220: The contour for t7 in the top view is inconsistent with the one in the cross section. Does the cross section go through the center of the plume?
We incorrectly said in the figure caption that the top view is always at the seeding height. However, it is at the height of the maximum ice crystal number concentration in each time step (plume). With the plume descending with time, also the maximum ice crystal number concentration is at lower heights. We adapted the figure caption as follows: "**Figure 5.** *Simulated ice crystal number concentration changes (cm⁻³, colormap) after seeding in simulation S26-2.5a with (a) showing the top view at the level of maximum concentration for each model output time step."* and the text as follows: *"Figure 5 shows the response in ice crystal number concentrations taken at the level of maximum concentration at each model output time step (top view) and as a cross section along the mean wind direction at each model output time step"*.

L228: "Note that the ice crystal number concentrations are spread ...": What is the difference between the ice crystal spreading out in this sentence and the previously described plume spreading out?
There is no difference, and we rephrased the sentences as follows (Line 252): *"The seeding plume not only spreads out horizontally, but also vertically due to turbulence and orographic lifting as shown in Fig. 5b, where we observe a vertical extent of up to 500 m."*.

L233: "from left to right and back": This is unclear; please clarify.
We rephrased the sentence as follows (Line 257): *"The radar performed repeated elevation scans in the plane perpendicular to the wind direction (from north-east), thus allowing us to observe the horizontal and vertical extent of the seeding signal."*.

L235: How is radar signal simulated? Did the authors use a radar emulator? Please describe.
Yes, we used a radar emulator, which is based on an existing diagnostic inside the model source code. This diagnostic is based on a Rayleigh approximation for the backscattering of the hydrometeors, where for frozen hydrometeors it differentiates between dry and wet ice,

snow, and graupel. The diagnostic takes the prognostic cloud masses into consideration and calculates the reflectivity.

L235: "a good qualitative agreement": The observations and simulations look quite different. It is unclear what features the authors referred to. Please clarify.
We adapted the text as follows (Line 259): *"While we can identify more fine-granular structures in the radar observation, the simulated radar reflectivity also shows an increase in reflectivity inside the seeding plume and a vertical spreading out throughout the cloud layer for the same time (10:30 UTC)."*.

L246: "the model configuration can be used to conduct seeding experiments for further investigation of the ice crystal growth inside the cloud": Based on Table 2, the median ice number concentration in the model is way lower than in the observation. Why? And what are the impacts of this discrepancy on the seeding experiments?
We have now also added the mean ice crystal number concentration to Table 2 and created a third table (Table 3) for Sect. 3.3.1, where we discuss the impact of the seeding particle concentration on ice crystal number concentration and cloud droplet reduction. We see that also the mean is underestimated for all 5 simulations. Only in the case of very high seeding particle concentrations ($10^7$ m$^{-3}$ s$^{-1}$) we find a higher mean than observed. The median is however always underestimated. We assume that this skewed distribution results from the quick spreading of the plume (turbulent diffusion in the model). This leads to few grid cells with high ice crystal number concentrations, and more grid cells with low concentrations. This bias towards lower concentrations could mean that the WBF is even more underestimated compared to observations, as a lower ice crystal number concentration implies less consumption of water vapor from evaporating cloud droplets.

L248: What is "the environment" here?
We rephrased the sentence as follows (Line 271): *"The seeding particle emission rate ($10^6$ seeding particles m$^{-3}$s$^{-1}$) used in this study is probably an upper estimate given that the surrounding of the seeding plume is in general warmer in the model (i.e., higher temperatures below the inversion) than observed which leads to a lower activation rate of INPs compared to reality."*.

L251: In this paragraph, the authors listed a few factors influencing the ice number concentration. How did they attribute the discrepancy or consistency between modeled and observed ice number concentration to one or all of these factors?
We cannot fully disentangle the different factors influencing the ice nucleating ability. We rephrased our hypothesis and also added that the freezing parameterization is constrained by the available cloud droplet number concentration. The model, however, underestimated the median cloud droplet number concentrations, which is now added in the supplement. The paragraph was now adapted to (Line 280): *"The model, however, fails to reproduce the very high concentrations of S25-2, which may be due to an underestimation of the ice nucleation activity of the seeding particles at temperatures close to -5 °C. Also, the ice crystal nucleation rate is constrained by the available cloud droplet number concentrations, which were underestimated in the model compared to the observations (Fig. B1). An additional reason could be the aerosol concentration, which was adapted to the simulation S26-2.5a. Hence, we cannot simulate the highest observed ice crystal number concentrations."*.

L255: "The model, however, fails to reproduce ...": Wasn't seeding rate tuned to match observed number concentrations?

We tuned the seeding concentration to match the experiment S26-2.5a on 26 January 2023, and used the same seeding particle emission rate for all simulations. We added the following sentence to Section 2.3 (Line 198): *"This seeding particle emission rate is based on a series of sensitivity simulations for seeding experiment S26-2.5a, where we injected different concentrations of seeding particles into the model and compared the simulated ice crystal number concentrations to the observations. The seeding particle emission rate and thus seeding setup were constrained by the ice crystal number concentrations observed by HOLIMO and tuned in such a way that they match for the seeding simulation S26-2.5a (Sect. 3.3.1)."*.

L256: "due to the aerosol concentration being adapted to the simulations S26-2.5a/b." Lost; please clarify

We tuned the seeding concentration to match the experiment S26-2.5a and now added this information in the methods section 2.3. We also removed the simulation S26-2.5b from the sentence above.

L278: "emphasizing the high efficiency of the WBF process": Is it possible to be secondary ice production?

We assume that the rate of secondary ice production is low as we only have very few larger cloud droplets (with radii > 20 $\mu$m) and riming only occurred in 2 of the 5 experiments (S26-2.5a/b). Given that the ice crystals only had a short time to grow (6-9 min) the splintering process occurs probably rarely, and if it does, these splinters did not grow large enough in that short amount of time to be detected in HOLIMO.

L284: "Both processes (WBF and riming) are parameterized in the model": Does WBF require a dedicated parameterization?

No, the WBF process is not directly parameterized but follows a vapor depositional growth rate equation, which checks the supersaturation with respect to ice for either growth or sublimation of the ice particles. Subsequently, the water vapor mass is adjusted. We added a more detailed description to the methods part of the model (Sect. 2.2).

L285: There are a few issues in this paragraph. First, the authors used Figure 9c as evidence that the model was not able to capture the observed cloud droplet depletion. However, in Figures 9a and 9b, there are no cloud droplet depletion and high ice number concentration around 10:30. What is the exact time period that the observations shown in Figure 9c come from? For this type of comparison, is the small sampling volume by HOLIMO suitable? Are the model data in Figure 9c from ice particle plume in the model domain? Please clarify. Is it possible that there is simply a mismatch between the observed and modeled arrival times? Second, "This discrepancy may originate ...": This is too much speculation. Did the authors perform any sensitivity simulations to prove? Are there any alternative explanations? Third, in Figures 9d and 9e, the authors commented that the model was performing well. Doesn't this suggest the WBF is doing fine? Why does this "further points to the fact that the WBF process in the model in its current form is not efficient enough"? Fourth, please show cloud water content/path, cloud droplet size distributions from observations and model for better comparison.

(1) We chose the model output time step closest to the expected seeding signal arrival time at the field site. The model data includes the entire plume, which covers several hundred meters in the horizontal and vertical dimensions (see Fig 5.). From Fig. 5 it is also notable that at the next time step (after 13 min from seeding start) the plume has already passed the field site. For better clarity, we added in the figure caption that the model output time steps correspond to the plumes in $t_2$, $t_4$, and $t_6$ in Figures 5 and 10.

(2) We have not yet done this sensitivity study, as this will be part of a next study where we also use a higher model resolution to investigate the role of turbulence, the shape factor and ventilation coefficient on ice crystal growth. We added a clarification to the sentence (Line 319): "*This discrepancy may originate from the computation of the ventilation coefficient, which determines the speeding up of the diffusional growth due to turbulent motions. This hypothesis will be investigated in future studies.*".

(3) We argue that only at later time steps the observed strong reductions in cloud droplets can be simulated, which points to the fact that the WBF process is slower in the model than observed.

(4) In Fig. 10, we show the cloud droplet and ice crystal size distributions for all model output time steps, and we see that the ice crystals grow over time, and cloud droplets shrink. However, at the observed time step ($t_2$), the model simulates a lower mean equivalent ice crystal radius than observed, while the cloud droplet radii are comparable.

L293: This paragraph is confusing. It seems that the authors were saying the model performs well regarding capturing observed cloud droplet reduction (starting from "Furthermore, ..." in L298), which contradicts previous conclusions.

We agree that the paragraph was confusing and adapted it as follows (Line 323): "*In Table 2, we show the median, mean, and maximum ice crystal number concentrations (absolute values). For cloud droplets, we report the reduction of the median, mean, and maximum cloud droplet number concentration relative to the undisturbed background (relative values) to account for the lower median cloud droplet number concentration of approximately 100 cm$^{-3}$ in the model compared to the observations (Fig. B1). The maximum ice crystal number concentrations are in good agreement (within ± 0.3 cm$^{-3}$) with observations in 4 out of 5 simulations. Only the S25-2 simulation strongly underestimates the maximum ice crystal number concentration by 1 cm$^{-3}$ (see Sect. 3.2.1 and Sect. 3.2.2), whereas the simulated median ice crystal number concentrations match the observations well. This is not the case for the other four simulations, where the median concentration is underestimated by an order of magnitude. When we also consider the mean values, we see that the model in general has only a few grid cells with high ice crystal number concentrations, while a lot of grid cells have very low ice crystal number concentrations. Regarding the changes in cloud droplets, the model fails to reproduce the maximum cloud droplet reductions, where 4 out of 5 simulations show almost no reduction. Only in the simulation S26-2.5a a stronger reduction in cloud droplet number concentration is notable. However, for all simulations the median and mean cloud droplet reductions are strongly underestimated.*".

L314: What is the difference between these two conditions? Did the authors mean (0 < w < w*) and (0 > w > w')?

Yes, this is correct. We forgot to add the lower/upper boundary of the regimes. For ice crystals, the updrafts are important, while for cloud droplets the downdrafts are important. We have adapted the formulas according to the reviewers' correction (Line 383).

L317: There are a few issues regarding the results in Figure 10. First, please clarify the data points going into the third and fourth rows (ice and liquid size distributions). In particular, is the integral of the area below each distribution the same as the total number of grid boxes in the plume? Second, due to the sedimentation of ice particles, the ice and liquid size distributions are for different particles and cannot be directly linked. Third, it seems that the main result is that at t2, (1) the observed ice particles are bigger than those in the simulations and (2) the observed liquid droplets are smaller. If one believes these two facts are linked, it is consistent with a weak WBF. But there could be many reasons that these two discrepancies are caused by different factors. Why can it be attributed to WBF? Fourth, if the main indicator of WBF is cloud droplet depletion, then all the data points contributing to the distributions in the fourth row are from grids that are less or not affected by WBF and these distributions do not support the argument anyway. Fifth, how do the liquid size distributions in the plume compare with background size distributions in the model? Sixth, the first two rows are interesting. Is there any dynamical factor that could lead to ice being too small and liquid too large? Like, does the vertical velocity distribution compare well with the observations? Does the release of the latent heat from seeding create its own circulation? This may affect the "background" cloud properties. There are some papers on this, for both marine stratocumuli seeded by ship emissions and mixed-phase clouds or supercooled liquid clouds seeded by ice, IIRC. Seventh, it seems that the distributions of the updraft vs the downdraft in the second row are inconsistent with those inferred from the first row (i.e., if one naively assumes the line between WBF_up and WBF_down separates the updraft and the downdraft). Is it simply because the vertical velocity in the cross section is not representative of the whole plume volume? Please clarify.

(1) We added in the figure captions that the radii calculations are based on the tracked plumes: "*Third row ((o)-(u)): Frequency distributions of equivalent ice crystal radius (μm, pink) over time and mean equivalent radius (downward facing triangle, pink numerical value) for the seeding plume at every model output time step.*".

(2) We further constrained the tracked plume in the simulation by the available cloud water mass which leads to a more focused investigation of the WBF process inside the cloud. This way we can assume that in the model and in the observations the seeding effect inside the cloud is compared.

(3) We would argue that the cloud droplets in the model and observations have similar mean radii, while the ice crystals show a larger discrepancy, which can be attributed to the WBF process.

(4) The grid cells going into the analysis of the fourth row are selected based on the plume with a ICNC threshold of 0.001 cm$^{-3}$. Hence, in all grid cells where we encounter cloud droplets, we also encounter ice crystals, which allows for the WBF process to take place.

(5) The cloud droplet size distribution for the background state of the model (see figure below) shows a higher mean size for all plume time steps. Hence, we have a reduction in cloud droplet size when we introduce ice crystals via seeding.

[Figure]

(6) 1st and 2nd row: Unfortunately, we do not have any observations on vertical velocities for the whole model domain. We cannot disentangle between the microphysics and dynamics influencing the growth of ice crystals / evaporation of cloud droplets. This is subject to future studies. The effect of the latent heat release from seeding is under debate. In Henneberger, Ramelli et al. (2023) it was discussed that some part of the updraft could be invigorated by the latent heat release.

(7) The vertical velocity cross sections are snapshots, and not averaged contour plots, that aim to highlight the dynamic structure of the cloud over time. In the WBF analysis (first row) all vertical velocities inside the plume were analyzed.

L334: Section 3.3.1: It is well-established that spherical ice particles do not grow fast enough, compared with ice particles with extreme habits. In the temperature range during the two days, the ice particles are likely to be needles/columns. Testing the effect of ice habits is probably more meaningful than increasing seeding rate by brute force.

It is true that spherical particles grow slower than other habits. However, in this study we wanted to investigate how the ice crystal growth in the current Swiss weather prediction model is represented. Additionally, various different modelling studies that have showed either a too strong or too weak WBF process, have all assumed spherical ice particles in the microphysics schemes (see Introduction). We do agree that the shape of the ice crystals is the next step to be investigated, and this will be done in future studies.

L343: "This sensitivity analysis also ...": Confused by this sentence; please clarify.

We adapted the sentence as follows (Line 380): "*Based on the results from this sensitivity analysis regarding the seeding particle emissions rate, we see that an emission rate of $10^6$ $m^{-3}s^{-1}$ is a good approximation to conduct seeding experiments in the model and to compare them to our observations in the field.*".

L345: "The default setup may still be ...": Confused by this sentence; please clarify.

This was wrong, thank you. We did not mean ice crystal number concentrations but seeding particles (Line 383): "*The seeding particle emission rate of $10^6$ $m^{-3}s^{-1}$ may still be an overestimation in seeding particles due to the warmer temperatures below the inversion in the model (see Fig. 3), which leads to lower ice nucleation rates of the seeding particles given the strong temperature dependence of ice nucleation.*".

L348: "Hence, if the model were colder, we would see a higher ice crystal number concentration.": This contradicts previous statement.

We made an error in the previous sentence and replaced now the ice crystal number concentrations by seeding particles (Line 383).

L360: "The observed seeding temperature was nevertheless eventually reached": This is unclear. How was it eventually reached?
We agree that it is written in an unclear way. We have adapted the sentence as follows (Line 398): *"The observed seeding temperature was nevertheless simulated, enabling us to conduct seeding simulations in the model."*.

L364: If the dilution for Jan. 25 cases (2 to 3 km) is so different from Jan. 26 cases (2.5 km), does it mean one cannot use Jan. 26 as a testbed for seeding on Jan. 25 and compare the results with Jan. 25 observations?
We adapted the conclusions based on our discussion in Sect. 3.2. We believe that the closest case (S25-2) is constrained both by the freezing parameterization at these temperatures and by the availability of cloud droplets, which limits the ice nucleating activity.

L367: "aerosol concentration": See comments for L256.
We tuned the seeding concentration to match the experiment S26-2.5a, and added now this information in the methods section 2.3 (Line 198). We also removed the simulation S26-2.5b from the sentence above.

TECHNICAL ISSUES:

Some of the comments below may reflect a personal preference in style. Feel free to ignore.

L24: Ice should be sublimating, not evaporating
We corrected this as follows (Line 24): *"(ii) cloud droplets evaporate and ice crystals sublimate ($e < e_{s,w}$) (Korolev, 2007)."*.

L24: "The second case ..." The sentence is not well-constructed. Please revise.
We rephrased the sentence as follows (Line 24): *"The case of ice crystals growing at the expense of cloud droplets (ii) is called the Wegener-Bergeron-Findeisen (WBF) process, which is caused by the difference in water vapor supersaturation between the liquid and ice phase (Wegener, 1911; Bergeron, 1935; Findeisen, 1938)."*.

L50: AgI does not consist of discrete molecules.
We adapted the sentence as follows (Line 53): *"… due to its lattice structure which closely resembles that of ice (DeMott, 1995; Marcolli et al., 2016)."*.

L316: "the serve": "they serve"?
Yes, we corrected this.

Figure 10: Maybe better to refer to the panels with two-digit labels, one for rows and one for columns. For example, b1 for second row and t1.
We follow here the guidelines of ACP.

L347: "stron temperature dependence": "strong"?
Yes, we corrected this.

---

## Author Comment (AC3)

**Authors' comments to Editor's comment to Anonymous Referee #1**

We selected below the comments highlighted by the editor and added the corresponding changes in the manuscript.
* * *
Line 113: "both experiments are identical in their setup". Since both seeding experiments are at the same location, is there any physical reason why there are two seeding experiments on that day? For example, I can understand S25-2, S25-2.5, S25-3 can test the impact of distance (growth time), but what about S26-2.5a and S26-2.5b? What can we learn from these two experiments?
In the field we conducted experiments using an identical setup to test the validity of the signal we observe in the radar and in-situ observations (similar to doing exact replicates in laboratory experiments). Here, we use both again to show that the model can reproduce the seeding signal consistently in the two experiments, but also when changing the seeding distance (other three experiments).
We added the following (line 118): "*These two identical setup serve to test the validity of the signal we observe in the field experiments, but also in the model simulations.*".

Line 149: "The frequency of model output was set to 5 min". The ice growth time is between 6 and 9 min (Table 1). Please comment on whether the relatively low output frequency would affect the comparison between observation and simulation.
In the table below we show the times of the expected arrival time and the closest model time step. The plume arrives at the field site almost always at a 5 min time step. We also tested 1 min output frequency for the simulation S26-2.5 and the results are similar. Hence, we follow here the 5 min output frequency. With the expected arrival time at the field site we also chose our model time step which may disagree with the time period of the observations given the differences / uncertainties in wind speed.

| Name | Seeding start (UTC) | Growth time (min) | Arrival at field site (UTC) | Closest model time step (UTC) |
|------|---------------------|-------------------|-----------------------------|-------------------------------|
| S26-2.5a | 10:22 | 8.0 | 10:30 | 10:30 |
| S26-2.5b | 10:48 | 7.1 | 10:55 | 10:55 |
| S25-2 | 10:50 | 6.1 | 10:56 | 10:55 |
| S25-2.5 | 10:28 | 8.0 | 10:36 | 10:35 |
| S25-3 | 11:15 | 9.1 | 11:24 | 11:25 |

We added following to the manuscript (line 156): "*The frequency of model output was set to 5 min after also testing1 min output frequency, which showed similar results as in the 5 min output. Moreover, calculating the expected arrival time of the seeding plume at the field site (seeding start and growth time, see Table 1) shows that the expected arrival and a full 5 min model output timestep are very close (within ± 1 min).*"

Please ensure that statements that observed and simulated ice number concentrations are in "good agreement" throughout the manuscript align with the revision that makes a more nuanced analysis; at least one earlier statement still states that they are in good agreement without such qualification.

Thank you, we have adapted this in the following (line 275): "*However, in both simulations, the median and mean concentrations are strongly underestimated, which is further discussed below (Sect. 3.3)*.".

---

## Author Comment (AC4)

**Authors' comments to Editor's comment to Anonymous Referee #2**

We selected below the comments highlighted by the editor and added the corresponding changes in the manuscript.
* * *
L65: Why "on the statistical significance"?
Given the natural variability in cloud systems, it is difficult to assess the feasibility of cloud seeding in an experimental setup. Here, numerical weather models can help to assess the uncertainty of cloud seeding by conducting repeated seeding simulations.
We added the following to the manuscript (line 68): "*Complementary to such field experiments, numerical models are employed to shed light on the statistical significance of cloud seeding by conducting repeated simulations in a controlled setup, which is not possible in a field experiment.*".

L187: "The seeding plume was defined ..." If the seeding plume is defined by a threshold, why did the authors still need the unseeded simulation as the background?
We need an unseeded simulation (reference simulation) to identify the changes in, e.g., cloud droplet number concentrations and vertical velocities for the grid cells affected by the seeding and not elsewhere in the model domain.
We added the following changes to the manuscript (line 209): "*We applied a simple method to extract the seeding signal from the background. We took the difference in ice crystal number concentrations between a seeding simulation and a reference simulation (no seeding) to remove the background and isolate the seeding plume. The seeding plume was then defined by a threshold ice crystal number concentration of 0.001 cm−3. We used the identified seeding plume as a mask for extracting further quantities in the seeding simulation, but also in the difference between the seeding and reference simulation, such as cloud droplet number concentrations, temperature, and updraft changes caused by the seeding perturbation.*".

L194: There are a few issues in Section 3.1. First, both Figures 3 and 4 showed that the temperature range where the seeding occurred was suitable for secondary ice production. Is this process parameterized in the model? Does it have an impact on the results? Second, the use of Jan. 26 as a testbed for Jan. 25 and then comparing the simulation results with Jan. 25 observations is questionable. Even though the seeding height is adjusted to match what actually occurred on Jan. 26, the meteorology for Jan. 26 seems to be different from Jan. 25 (cloud temperature range, thermodynamic profiles, maybe also liquid water content profile, etc.) Please carefully justify this decision.
Secondary ice production is simulated in the model following Hallet and Mossop. However, it only occurs if graupel or hail particles are rimed, and then a splintering rate is calculated. In our simulations the amount of graupel particles is close to 0, so we do not see an effect of SIP.
We added following to the manuscript (line 176): "*At subzero temperatures secondary ice production can occur, which is also parameterized in the model. For secondary ice production to occur in the model rimed graupel particles are needed, but their concentrations are close to zero in the model; hence we can exclude the effect of secondary ice production in our analysis*".

Given that the model simulation from 26 January 2023 reproduces the cold temperatures from 25 January 2023, without the sharp inversion at cloud top, we believe it is adequate to use the 26 January 2023 simulation as a testbed. While we do have a lower cloud on 25 January 2023, we still encounter a persistent low stratus cloud with north-easterly to easterly winds as shown in Table 1. The wind speeds are comparable in the observations and the model, and this further supports our method to use 26 January 2023 as a surrogate model.

We added the following to the manuscript (line 182): "*The selection of the presented seeding simulations was constrained by how accurately the model reproduced the observed environmental conditions. Unfortunately, the model overestimated the temperatures for 25 January 2023 (Henneberger et al., 2023) (Fig. 3a), while the temperatures on the 26 January 2023 were simulated adequately (Fig. 3b). For this reason, we decided to utilize the simulation of 26 January 2023 for all seeding experiments conducted on 25 and 26 January 2023 (see also Sect. 3.1) given the presence of persistent low stratus clouds with north-easterly to easterly winds on both days.*".

L209: "predicted cloud cover": How did the authors define cloud cover from radar observations and simulations?

The reflectivity of the cloud radar (FMCW-94-DP, Radiometer Physics GmbH) is used as a proxy for cloud cover observed at the field site. In the model, the cloud cover is diagnosed based on the prognostic cloud water mass.

We added the following to the manuscript (line 243): "*In addition, we compared the observed and predicted cloud cover at the field site by taking the radar reflectivity of a vertically pointing radar as a proxy for cloud cover and the computed cloud cover from the prognostic cloud water mass in the model (Fig. 4).*".

L235: How is radar signal simulated? Did the authors use a radar emulator? Please describe.

Yes, we used a radar emulator, which is based on an existing diagnostic inside the model source code. This diagnostic is based on a Rayleigh approximation for the backscattering of the hydrometeors, where for frozen hydrometeors it differentiates between dry and wet ice, snow, and graupel. The diagnostic takes the prognostic cloud masses into consideration and calculates the reflectivity.

We added the following to the manuscript (line 270): "*The simulated radar reflectivity is based on an implemented Rayleigh approximation for the backscattering of the cloud particles, where for frozen hydrometeors it is differentiated between dry and wet ice, snow, and graupel.*".

L278: "emphasizing the high efficiency of the WBF process": Is it possible to be secondary ice production?

We assume that the rate of secondary ice production is low as we only have very few larger cloud droplets (with radii > 20 $\mu$m) and riming only occurred in 2 of the 5 experiments (S26-2.5a/b). Given that the ice crystals only had a short time to grow (6-9 min) the splintering process occurs probably rarely, and if it does, these splinters did not grow large enough in that short amount of time to be detected in HOLIMO.

We added the following to the manuscript (line 178): "*During the field experiments, we also expect a low secondary ice production rate given that only a few larger cloud droplets (with radii > 20 µm) are present. Riming on the ice crystals was also only visible in two out of the*

*five experiments (S26-2.5a/b). In addition, if splinters occurred, they probably did not grow large enough to be detected given the short growth time during the experiments (see Table 1).".*

L317: There are a few issues regarding the results in Figure 10. First, please clarify the data points going into the third and fourth rows (ice and liquid size distributions). In particular, is the integral of the area below each distribution the same as the total number of grid boxes in the plume? Second, due to the sedimentation of ice particles, the ice and liquid size distributions are for different particles and cannot be directly linked. Third, it seems that the main result is that at t2, (1) the observed ice particles are bigger than those in the simulations and (2) the observed liquid droplets are smaller. If one believes these two facts are linked, it is consistent with a weak WBF. But there could be many reasons that these two discrepancies are caused by different factors. Why can it be attributed to WBF? Fourth, if the main indicator of WBF is cloud droplet depletion, then all the data points contributing to the distributions in the fourth row are from grids that are less or not affected by WBF and these distributions do not support the argument anyway. Fifth, how do the liquid size distributions in the plume compare with background size distributions in the model? Sixth, the first two rows are interesting. Is there any dynamical factor that could lead to ice being too small and liquid too large? Like, does the vertical velocity distribution compare well with the observations? Does the release of the latent heat from seeding create its own circulation? This may affect the "background" cloud properties. There are some papers on this, for both marine stratocumuli seeded by ship emissions and mixed-phase clouds or supercooled liquid clouds seeded by ice, IIRC. Seventh, it seems that the distributions of the updraft vs the downdraft in the second row are inconsistent with those inferred from the first row (i.e., if one naively assumes the line between WBF_up and WBF_down separates the updraft and the downdraft). Is it simply because the vertical velocity in the cross section is not representative of the whole plume volume? Please clarify.

(1) We added in the figure captions that the radii calculations are based on the tracked plumes: *"Third row ((o)-(u)): Frequency distributions of equivalent ice crystal radius (μm, pink) over time and mean equivalent radius (downward facing triangle, pink numerical value) for the seeding plume at every model output time step."*.

(2) We further constrained the tracked plume in the simulation by the available cloud water mass which leads to a more focused investigation of the WBF process inside the cloud. This way we can assume that in the model and in the observations the seeding effect inside the cloud is compared. We added the following in the manuscript (line 330): *"In addition, we constrained the seeding plume by the available liquid water content inside the cloud: We only considered grid cells in the analysis where the ice crystal number concentration is larger than 0.001 cm$^{-3}$ and the liquid water content larger than 0.1 gm$^{-3}$. This way we only include grid cells where the WBF process actually could take place."*.

(3) We would argue that the cloud droplets in the model and observations have similar mean radii, while the ice crystals show a larger discrepancy, which can be attributed to the WBF process.

(4) The grid cells going into the analysis of the fourth row are selected based on the plume with a ICNC threshold of 0.001 cm$^{-3}$. Hence, in all grid cells where we encounter cloud droplets, we also encounter ice crystals, which allows for the WBF process to take place. See also answer to point 2.

(5) The cloud droplet size distribution for the background state of the model (see figure below) shows a higher mean size for all plume time steps. Hence, we have a reduction in cloud droplet size when we introduce ice crystals via seeding. We added this figure to the appendix and reference it here in the text (line 375): "*Also the mean radius of cloud droplets in the reference simulation is consistently larger than in the seeding simulation (see Fig. D1).*".

[Figure]

(6) 1st and 2nd row: Unfortunately, we do not have any observations on vertical velocities for the whole model domain. We cannot disentangle between the microphysics and dynamics influencing the growth of ice crystals / evaporation of cloud droplets. This is subject to future studies. The effect of the latent heat release from seeding is under debate. In Henneberger, Ramelli et al. (2023) it was discussed that some part of the updraft could be invigorated by the latent heat release. We added the following to the manuscript (line 372): "*We note here, that we cannot distinguish between the microphysical (latent heat release) and dynamical (topography and wind field) influence on ice crystal growth and evaporation of cloud droplets. Henneberger et al. (2023) discussed that some updraft invigoration may occur due to latent heat release upon ice nucleation, however this is still under debate.*".

*(7) The vertical velocity cross sections are snapshots, and not averaged contour plots, that aim to highlight the dynamic structure of the cloud over time. In the WBF analysis (first row) all vertical velocities inside the plume were analyzed. We added the following to the figure caption: "**Figure 10**: [...] Second row ((h)-(n)): Cross sections of vertical velocity (instant values) along the mean wind direction over time.[...]"*

L334: Section 3.3.1: It is well-established that spherical ice particles do not grow fast enough, compared with ice particles with extreme habits. In the temperature range during the two days, the ice particles are likely to be needles/columns. Testing the effect of ice habits is probably more meaningful than increasing seeding rate by brute force.

It is true that spherical particles grow slower than other habits. However, in this study we wanted to investigate how the ice crystal growth in the current Swiss weather prediction model is represented. Additionally, various different modelling studies that have showed either a too strong or too weak WBF process, have all assumed spherical ice particles in the microphysics schemes (see Introduction). We do agree that the shape of the ice crystals is the next step to be investigated, and this will be done in future studies.

We added the following in the manuscript (line 168): "*The ice crystal shape is set to be spherical, which is a simplification in the scheme given the large variety of shapes (Bailey and Hallett, 2009). In this study, we do not change the shape of the ice crystals as we want to investigate the ice crystal growth rate in the default configuration of the model. During the conducted seeding experiments, we mostly measured needles or columns. When we compare the ice crystal sizes in Fig. 10, we investigate the mean equivalent radius of ice crystals.*".

Within the manuscript text, please more directly address the reviewer concern regarding potential mismatch between the observed and modeled arrival times (or indicate where that has been done).

We added following to the manuscript (line 156): *"The frequency of model output was set to 5 min after also testing1 min output frequency, which showed similar results as in the 5 min output. Moreover, calculating the expected arrival time of the seeding plume at the field site (seeding start and growth time, see Table 1) shows that the expected arrival and a full 5 min model output timestep are very close (within ± 1 min)."*